# Biomass-burning and urban emission impacts in the Andes Cordillera region based on in-situ measurements from the Chacaltaya observatory, Bolivia (5240 m a.s.l.)

Chauvigné Aurélien[1,11], Diego Aliaga[2], Karine Sellegri[1], Nadège Montoux[1], Radovan Krejci[3], Griša Močnik[4], Isabel Moreno[2], Thomas Müller[5], Marco Pandolfi[6], Fernando Velarde[2], Kay Weinhold[5], Patrick Ginot[7], Alfred Wiedensohler[5], Marcos Andrade[2,8], Paolo Laj[7,9,10]

[1] Laboratoire de Météorologie Physique, OPGC, CNRS UMR6016, Université d'Auvergne, Clermont-Ferrand, France

[2] Laboratorio de Física de la Atmósfera, Universidad Mayor de San Andrés, La Paz, Bolivia

[3] Department of Environmental Science and Analytical Chemistry & Bolin Centre of Climate Research, Stockholm University, Stockholm 10691, Sweden

[4] Condensed Matter Physics Department, Jožef Stefan Institute, Ljubljana, Slovenia

[5] Leibniz Institute for Tropospheric Research, Permoserstr. 15, 04318 Leipzig, Germany

[6] Institute of Environmental Assessment and Water Research, c/ Jordi-Girona 18-26, 08034, Barcelona, Spain

[7] Univ-Grenoble-Alpes, CNRS, IRD, Grenoble-INP, IGE, 38000 Grenoble, France

[8] Department of Atmospheric and Oceanic Science, University of Maryland, College Park, MD, USA

[9] CNR-ISAC, National Research Council of Italy – Institute of Atmospheric Sciences and Climate, Bologna, Italy

[10] University of Helsinki, Atmospheric Science division, Helsinki, Finland

[11] Now at : Laboratoire d'Optique Atmosphérique, CNRS UMR8518, Université de Lille, Lille, France

*Correspondence to:* Aurélien Chauvigné (aurelien.chauvigne@univ-lille.fr)

**Abstract.** This study documents and analyzes a 4-year continuous record of aerosol optical properties measured at the Global Atmosphere Watch (GAW) station of Chacaltaya (5240 m a.s.l.), in Bolivia. Records of Particle light scattering and particle light absorption coefficients are used to investigate how the high Andean cordillera is affected by both long-range transport and by the fast-growing agglomeration of La Paz / El Alto, located approximately 20 km away and 1.5 km below the sampling site. The extended multi-year allows to study the properties of aerosol particle for different air-mass types, during wet and dry seasons, also covers periods with the site affected by biomass-burning in the Bolivian lowlands and the Amazonian basin. The absorption, scattering and extinction coefficients (median annual values of 0.74, 12.14 and 12.96 $Mm^{-1}$ respectively) show a clear seasonal variation with low values during the wet season (0.57, 7.94 and 8.68 $Mm^{-1}$ respectively) and higher values during the dry season (0.80, 11.23 and 14.51 $Mm^{-1}$ respectively). The record is driven by variability at both seasonal and diurnal scales. At diurnal scale, all records of intensive and extensive aerosol properties show a pronounced variation (daytime maximum, nighttime minimum), as a result of the dynamic and convective effects. The particle light absorption, scattering and extinction coefficients are on average 1.94, 1.49 and 1.55 times higher, respectively, in the turbulent thermally driven conditions as respect to the more stable condition, due to more efficient transport from the boundary layer. Retrieved intensive optical properties are significantly different from one season to the other, reflecting the changing aerosol emission sources of aerosol at larger scale. Using wavelength dependence of aerosol particle optical properties, we discriminated contributions from natural (mainly mineral dust) and anthropogenic (mainly biomass-burning and urban transport or industries) emissions according to seasons and local circulation. The main sources influencing measurements at CHC are from the urban area of La Paz/El Alto in the Altiplano, and regional biomass-burning from the Amazonian basin. Results show a 28% to 80% increase in the extinction coefficients during the biomass-burning season with respect to the dry season, which is observed in both tropospheric dynamic conditions. From this analysis, long-term observations at CHC provides the first direct evidence of the impact of Biomass Burning emissions of the Amazonian basin and urban emissions from La Paz area on atmospheric optical properties to a remote site all the way to the free troposphere.

## Introduction:

Natural and anthropogenic aerosol particle emissions significantly influence the global and regional climate by absorbing and scattering solar radiation (Charlson et al., 1992; Boucher et al., 2013; Kuniyal et Guleria 2018). Global-scale estimates of aerosol radiative forcing are still highly uncertain. Regional and local scale radiative forcing estimates show large variability, reflecting the dependence on highly variable factors such as the ground albedo, aerosol particle loadings, as well as the nature and localization of the aerosol particle in the atmosphere. While aerosol particles have a net cooling effect at a global-scale ($-0.35$ W.m$^{-2}$, Myrhe et al., 2013), the sign of local direct radiative forcing is determined by a balance between cooling by most aerosol species (sulfates, nitrates, organic aerosols and secondary organic aerosols), and warming by black carbon (BC) that absorbs solar radiation (Myhre et al., 2013).

The major sources of BC particles are biomass-burning and incomplete fuel combustion. The Amazonian Basin accounts for approximatively fifty percent of the global tropical forest area and shrink more than 2% every year, which makes it one of the most important sources of BC particles. However, long term measurements at high altitude are still poorly documented in this region. Some observations of regional aerosol burden show the intense emission sources for both primary and secondary aerosol particle and their local impacts. Martin et al. (2010) report evidence of natural and anthropogenic emissions in this region with clear seasonal variations of atmospheric particle concentrations close to the surface. Artaxo et al. (2013) retrieved from a sampling location close to nearby recurrent fires (close to Porto Velho) concentrations of biomass particles 10 times higher during the dry period than during wet period. The authors also report that the particles concentrations at this site are 5 times higher than at a remote site in the same area, and both sites show a clear seasonal variation.

In addition to the aforementioned studies on the aerosol burdens, several other studies show important modifications of the atmospheric optical properties during biomass-burning (BB) episodes, in the Amazonian basin and in la Plata basin at the end of the dry season (August - September). Between the wet season and the biomass burning season, Schafer et al. (2008) show an increase of Aerosol Optical Depth by a factor of 10 from AERONET sites in southern forest region and the Cerrado region and, by a factor of 4 in the northern forest region. Husar et al. (2000) have reported extinction coefficients on the integrated visible range (or visibility) in the Amazon basin at four different altitude stations during the BB period. The study reports a spatial pattern of the visibility between 100 and 200 Mm$^{-1}$ over the Amazon Basin. However, values can reach 600 Mm$^{-1}$ at Sucre station (2903 m above sea level, hereafter abbreviated as "a.s.l."), 1000 Mm$^{-1}$ at Vallegrande (1998m a.s.l.) and 2000 Mm$^{-1}$ at Camiri (792 m a.s.l.) during BB period. Even the study clearly shows impacts of Amazonian activities at different altitudes and long distances, only few studies report long time period of aerosol optical properties. At "Fazenda Nossa Senhora Aparecida" (FNSA) station in Brazil (770 m a.s.l.), Chand et al. (2006) report the absorption and scattering coefficients reaching 70 Mm$^{-1}$ (at 532 nm) and 1435 Mm$^{-1}$ (at 545 nm) respectively during large-scale BB events (PM2.5 > 225µg.m$^{-3}$) from ground-based measurements. These extremely high coefficients are due to the proximity to BB sources for FNSA station and its very low altitude.

Only a few studies report BC transport through different atmospheric layers. During the Large-Scale Biosphere-Atmosphere Experiment in Amazonia (LBA, in March 1998), Krejci et al. (2003) retrieved particle concentrations in the free Tropospheric Layer (above 4 km a.s.l.), 2 and 15 times higher than in the boundary layer, due to new particle formation in BB plumes. From airborne LIDAR measurements during SAMBBA, Marenco et al. (2016) have also observed high particle concentrations at high altitude (between 1 to 6 km). Their work highlight long range transport of biomass-burning plumes with lifetimes of several weeks. Chand et al. (2006) also demonstrate increasing particle scattering with altitude, partly explained by particle coagulation and condensation of gases during transport. Bourgeois et al. (2015) show from satellite remote sensing measurements (Cloud-Aerosol Lidar with Orthogonal Polarization, CALIOP), that BB particles originating from the Amazonian Basin reach the altitude of 5 km a.s.l. Contrary to Krejci et al. (2003) and Chand et al. (2006), they show a constant decrease of aerosol particle extinction with altitude, at a rate of 20 Mm$^{-1}$ per kilometer of altitude. Hamburger et al. (2013) present long term (3 years) ground-based measurements at Pico Espejo (4765 m a.s.l.), Venezuela. They show the

influences of the local Venezuelan savannah and of the Amazonian basin biomass Burning emissions, mainly during the dry period, and into the whole Tropospheric layer.

A challenging part is to separate the contributions of different aerosol sources from retrieved optical properties. For example, the single scattering albedo (SSA) is closely related to the particle size, and determines the magnitude of the aerosol radiative forcing (Hansen et al., 1997). For Tropical BB events, the SSA is around 0.83 at 550 nm for a fresh plume and increases with time up to 0.87 (Reid et al., 2005). On the other hand, the spectral dependency of aerosol optical properties, the Angström exponents, decrease during the aging process of the smoke. From five different campaigns over different continents using AERONET sites, Russel et al. (2010) work permits to define thresholds on the absorption and scattering Angström exponents (respectively AAE and SAE) for urban pollution, BB and dust particles. They associated urban pollution particles to AAE close to 1, whereas BB particles to AAE close to 2. In addition, SAE values are close to 1 for Dust particles and close to 2 for urban particles. Similar results are observed from Clarke et al. (2007) work, based on in-situ airborne measurements over North-America. Their work show BB plumes with AAE values close to 2,1 and polluted plume with AAE close to 1. The correlation between the single scattering albedo Angström exponent (SSAAE) and the concentration of dust allows to define that air masses with SSAAE values below 0 are mainly influenced by dust sources, contrarily to urban pollution sources that show values above 0 (Collaud Coen et al., 2004). This has also been confirmed from different AERONET sites in the world (Dubovik et al., 2002).

| Aerosol type | SAE | AAE | SSAAE |
|---|---|---|---|
| Dust | Close to 1 | Close to 1 | Below 0 |
| Urban pollution | Close to 2 | Close to 1 | Higher than 0 |
| Biomass burning | | Close to 2,1 | |

**Table 1: Expected aerosol type and their optical properties for each cluster according season and atmospheric stability.**

As a summary, Table 1 shows expected Angström exponent for dust, urban pollution and Biomass Burning particles according the different referenced works (Dubovik et al., 2002 ; Collaud Coen et al., 2004 ; Clarke et al., 2007 ; Russel et al., 2010). This information has to be taken with caution since source influences are expected homogeneous and have been reported from several regions. The present work aims at evaluating the contribution of anthropogenic and natural particle to the global optical properties of aerosols measured at a high altitude background site at Chacaltaya (Bolivia) over a four-year period (2012-2015). Monthly and diurnal variations of extensive optical properties (related to particle concentration) and intensive optical properties (related to particle chemistry) are firstly shown. A robust method based on the measurement of the atmospheric stability is then applied to distinguish atmospheric conditions (stable and turbulent). Finally, back-trajectory analysis and optical wavelength dependences are presented to identify impacts of local and regional aerosol sources.

## 1. Site description

The urban area of La Paz/El Alto, extends from approximately 3200 m to more than 4000 m a.s.l. in the Altiplano. It is a fast growing urban area with a population of c.a. 1.7 million inhabitants, covering a complex topography. In this region, meteorological conditions are governed by wet and dry seasons. The wet season span from December to March, and the dry season from May to September. April, October and November are considered as transition periods between the two main seasons. Between May and October, agricultural practices in the Yungas (closest valleys to the La Paz plateau and the Amazonian Basin) and the Amazon and La Plata basins include intense vegetation burning (Carmona-Moreno et al. 2005, Giglio et. al 2013). Indeed, the closest region where large areas are affected by biomass burning activities is the Bolivian Amazonia (Beni, Santa Cruz, north of La Paz departments) located ca. 300 km from the station, north and eastward from the Andes mountain range. In this paper, were choose to define August and September as the biomass-burning (BB) period because those are the months when the BB activities are more intense in the aforementioned region.

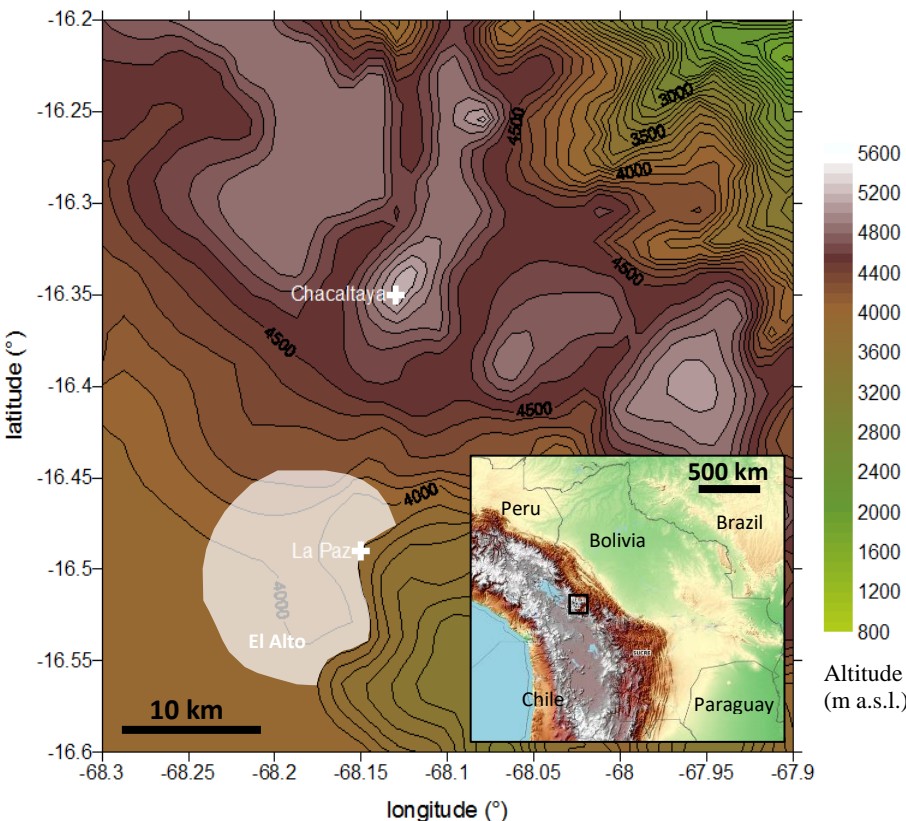

**Figure 1: Topographic description of La Paz and Chacaltaya region, and Bolivia in the lower right panel. The black rectangle on the small panel represents La Paz region. The urban area of La Paz-El Alto (marked as white shading) lies in the Altiplano high-plateau at around 4000 m a.s.l..**

The in-situ measurement site used in this study is the high altitude station of Chacaltaya GAW (Global Atmospheric Watch) (site code: CHC, and coordinates: 16°21 S, 68°07 W), located at 5.240 m a.s.l., at 17 km North of La Paz, as shown in Fig. (1). In-situ instruments of the station operated behind a Whole Air Inlet equipped with an automatic dryer (activated above 90% RH). The station continuously measures concentrations of trace gases, and physical and chemical properties of aerosols since 2011 (Rose et al., 2015). In the current study, the full dataset of in-situ optical measurements has been used between January 2012 and December 2015.

## 2. Instrument and methods

### 2.1. In-situ measurements

Absorption and scattering coefficients of the aerosol were measured in dry conditions (< 40 %) using an Aethalometer (Magee Scientific AE31) at 7 different wavelengths (370, 470, 520, 590, 660, 880 and 950 nm), a

Multi-Angle Absorption Photometer (MAAP, Thermo Scientific) at 635 nm and an integrated Nephelometer (Ecotech Aurora 3000) at 3 wavelengths (450, 525 and 635 nm). The Aethalometer measures the rate of change of optical transmission of the filter on which particles are collected at 5-minute resolution. Every 5 minutes, the spot on the filter band is changed in order to reduce loading effects. The reference of the transmissivity is the part of the same filter without particles (Hansen et al., 1982). Sensor calibration is performed automatically and an uncertainty of 5 % on attenuation coefficients is given by the constructor. Aethalometer measurements were compensated for multi-scattering effects and loading effects (or shadowing effects) following the method described by Weingartner et al. (2003) briefly explained below. As described by Weingartner et al. (2003) and used on Chacaltaya measurements by Rose et al. (2015), the absorption coefficient $\sigma_{abs}$ is retrieved from BC concentrations measured from the aethalometer. From BC concentrations at every measurement spots, attenuation coefficients $\sigma_{atn}$ at different wavelengths are retrieved as:

$$\sigma_{atn}(\lambda) = BC(\lambda) \cdot \sigma_m(\lambda) \tag{1}$$

with $\sigma_m$ the mass coefficients given by the instrument's instructions (The Aethalometer, A.D.A. Hansen, Magee Scientific Company, Berkley, California, USA) and based on the Mie theory. $\sigma_m$ strongly depends on the aerosol type and age (from 5 to 20 m² g$^{-1}$, Liousse et al., 1993). However, the manufacturer values (14625 nm m$^2$ g$^{-1}$ $\lambda^{-1}$) have been recently validated in a comparison study between different aethalometer corrections (Collaud Coen et al., 2010 ; Saturno et al., 2017).

The absorption coefficient is then calculated with the following equation:

$$\sigma_{abs}(\lambda) = \frac{\sigma_{atn}(\lambda)}{C \cdot R(\lambda)} \tag{2}$$

with $C = 3.5$ a calibration factor linked to multiple-scattering and assumed constant according wavelengths (GAW Report No. 227),

and R, a calibration factor which depends on aerosol loading on the filter and aerosol optical properties, calculated as:

$$R(\lambda) = \left(\frac{1}{f(\lambda)} - 1\right) \frac{\ln(\sigma_{atn}(\lambda)) - \ln(10\%)}{\ln(50\%) - \ln(10\%)} + 1 \tag{3}$$

where f is the filter loading effect compensation parameter and represents the slope of the curve of R as function of $\ln(\sigma_{atn})$ for a $\sigma_{atn}$ change from 10% to 50% . This factor is adjusted to obtain a median ratio between the absorption coefficient before and after spot changes close to 1.

Similarly to the Aethalometer, the MAAP measures the radiation transmitted and scattered back from a particle-loaded fiber filter (Petzold and Schönlinner, 2004). According to Petzold and Schönlinner (2004), uncertainty of the absorbance is 12%. A mass absorption cross-section $Q_{EBC} = 6.6$ m$^2$.g$^{-1}$ at 670 nm is used to determine Equivalent Black Carbon mass concentrations ($m_{EBC}$) from absorption coefficient ($\sigma_{abs}$) and a wavelength correction factor of 1.05 was applied according Equ. (4) to obtain $\sigma_{abs}$ at 635 nm (Müller et al., 2011a) from measurement at 637 nm.

$$\sigma_{abs} = 1.05 \, m_{EBC} \, Q_{EBC} \tag{4}$$

The nephelometer measures the integrated light scattered by particles. Because the angular integration is only partial (from 10° to 171°), nephelometer data were corrected for truncation errors, but also for detection limits according to Müller et al. (2011b). The instrument permits to retrieve aerosol particle scattering coefficients ($\sigma_{scat}$ from 10° to 171°). The nephelometer instrument is calibrated using CO2 as span gas and frequent zero adjusts were performed, following the procedure described in Ecotech manual (2009). The uncertainty of the Aurora 3000 is given in the user manual to be 2,5 %. However, it has been noticed that the three wavelengths of the Chacaltaya's nephelometer do not present equivalent robustness. Indeed, measurements at 635 nm remain unstable during the analyzed period and are thus not selected for the following results.

More optical parameters can be retrieved from the combination of these instruments. The extinction coefficient ($\sigma_{ext}$) and aerosol particle single scattering albedo (SSA) are calculated according to Equ. (5) and (6). In addition,

the full spectral information of each instrument is fitted by a power-law (Equ. (7) and (8)) and allow to retrieve aerosol particle Angström exponents such as the scattering Angström exponent (SAE) from nephelometer measurements, the absorption Angström exponent (AAE) from aethalometer measurements and the single scattering albedo Angström exponent (SSAAE).

$$\sigma_{ext}(\lambda) = \sigma_{abs}(\lambda) + \sigma_{scat}(\lambda) \tag{5}$$

$$SSA(\lambda) = \frac{\sigma_{scat}(\lambda)}{\sigma_{scat}(\lambda) + \sigma_{abs}(\lambda)} \tag{6}$$

$$\sigma_{abs}(\lambda) = b_{abs} \times \lambda^{-AAE} \tag{7}$$

$$\sigma_{scat}(\lambda) = b_{scat} \times \lambda^{-SAE} \tag{8}$$

$$SSA(\lambda) = b_{ssa} \times \lambda^{-SSAAE} \tag{9}$$

With $b_{abs}$, $b_{scat}$ and $b_{ssa}$, and AAE, SAE and SSAAE the power-law fit coefficients.

### 2.2. Method for differentiating stable or turbulent conditions at CHC

As is often the case for mountain sites, CHC is strongly influenced by thermal circulation, developed on a daily basis on mountain slopes (Whiteman, 2000). Depending on the time of the day and the season, the CHC high-altitude site can be influenced by air-masses from the mixing layer, the residual layer or the lower free troposphere. The mixing layer height is driven by convective processes related to surface temperature, with higher mixing layer height during daytime and lower height during nighttime. In addition to the diurnal mixing layer cycle, the complex mountainous topography of the area affects local circulation by channeling the air flow, thus complicating the differentiation between the mixing layer and free troposphere. In addition, a residual layer can also be present at CHC station during nightime, resulting from low dispersion of the daytime convection. Because no clear distinctions between the mixing, the free tropospheric, and the residual layers can be strictly obtained from in-situ measurements only, the present dataset recorded at Chacaltaya station is separated in terms of stability conditions (turbulent and stable). To differentiate stable conditions (SC; typically the free Tropospheric layer, but also RL) from turbulent conditions (TC; typically the mixing layer, but also cloudiness over the station or wind channelling effects), we used a methodology described in Rose et al. (2017). This method is based on the hourly averaged value of the standard deviation of the horizontal wind direction ($\sigma_\theta$ in Eq. 10) calculated every 15 minutes:

$$\sigma^2_{\theta(1h)} = \frac{\sigma^2_{\theta(15)} + \sigma^2_{\theta(30)} + \sigma^2_{\theta(45)} + \sigma^2_{\theta(60)}}{4} \tag{10}$$

with $\sigma_{\theta(15)}$ the standard deviation of the horizontal wind direction calculated on the first 15 minutes of every hour, and $\sigma_{\theta(60)}$ the last 15 minutes of every hour.

$$\sigma_\theta = \sin^{-1}(\varepsilon)[1.0 + b\varepsilon^3] \tag{11}$$

and $b = 2/\sqrt{3} - 1 = 0.1547$, $\varepsilon = \sqrt{1 - (s_a^2 + c_a^2)}$

with the averages $s_a = \frac{1}{N}\sum_{i=1}^{N} sin\theta_i$ and $s_a = \frac{1}{N}\sum_{i=1}^{N} cos\theta_i$ of N the number of horitontal wind direction ($\theta_i$) recorded in 15 minutes.

A smoothed threshold is used to separate TC and SC ranges from 12.5° to 18° for the dry season and from 12.5° to 22.5° for the wet season based on Mitchell (1982)'s recommendations and on BC analyses (Rose et al., 2017). Interface cases correspond to unclassified data which mainly show high variability of the standard deviation between the two categories of dynamic. As described in Rose et al. (2017), the classification depends also on the $\sigma_\theta$ value in the 4-hour time interval across the time of interest. Interface cases correspond to unclassified data

which mainly show a high variability of the standard deviation between the two categories of dynamic. For clarity, the interface cases are excluded from the dataset in the rest of the paper.

The standard deviation of horizontal wind direction at CHC highlights the diurnal cycle between stable and turbulent conditions directly related to temperature and the behavior of the atmospheric boundary layer (ABL). This influence of the TC at CHC is due to its particular topographical setting, particularly due to its proximity to the Altiplano plateau (altitude > 3 km, 200 km width near CHC). This high and semiarid plateau receives significant amounts of solar radiation that heat the surface, producing an expansion of the TC as observed in Lidar measurements near the station (Wiedensohler et al. 2018).

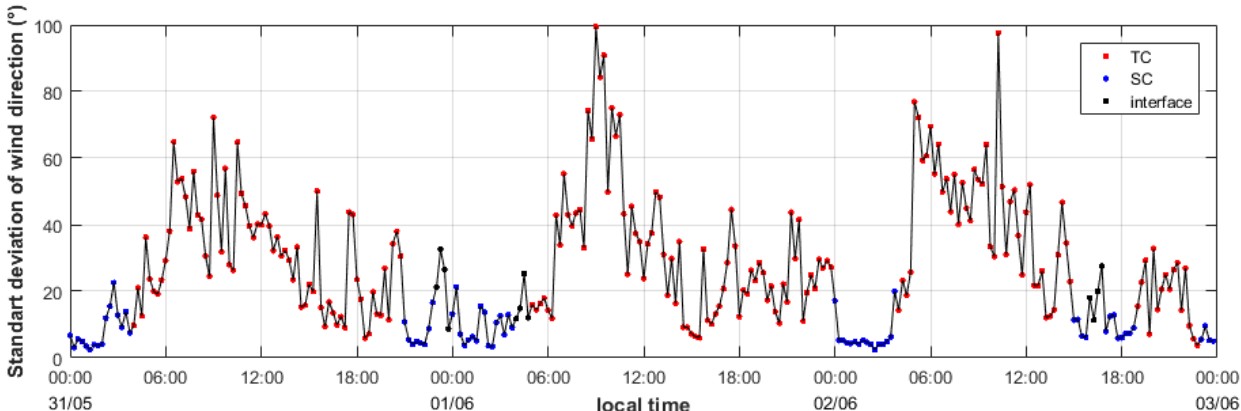

**Figure 2: Standard deviation of the wind direction measured at the high altitude site of Chacaltaya (5240 m a.s.l.) from the 31st of May to the 2nd of June 2012 at 1-hour resolution. Red points corresponding to turbulent condition (TC) cases, blue points to stable condition (SC) cases and black points to undefined/interface cases. Time corresponds to the local time.**

Figure (2) shows the standard deviation of the horizontal wind direction during a 3-day period (from 31 May 2015 to 2 June 2012) with blue lines representing SC cases and red lines, TC cases. Black spots represent undefined cases (or interface) due to a fluctuating classification within the 1-hour time window. This 3-day example shew that SC are mostly observed during night when the convective effect of the previous day is already dissipated and no convective effect of the current day is present.

Following this classification, the average monthly and diurnal variation of the fraction of SC, TC and undefined conditions for each season for 4 years of measurements (from 2012 to 2015) were calculated and are represented in Fig. (3). For each season, SC are dominant before 10:00 (local time) and after 18:00 whereas TC are mostly observed during daytime. This tendency is mostly observed during the dry season with more than 60% of TC in daytime and 80% of SC in nighttime. However, monthly variations show similar tendencies during the full year with around 60% of time in the stable condition (SC in blue).

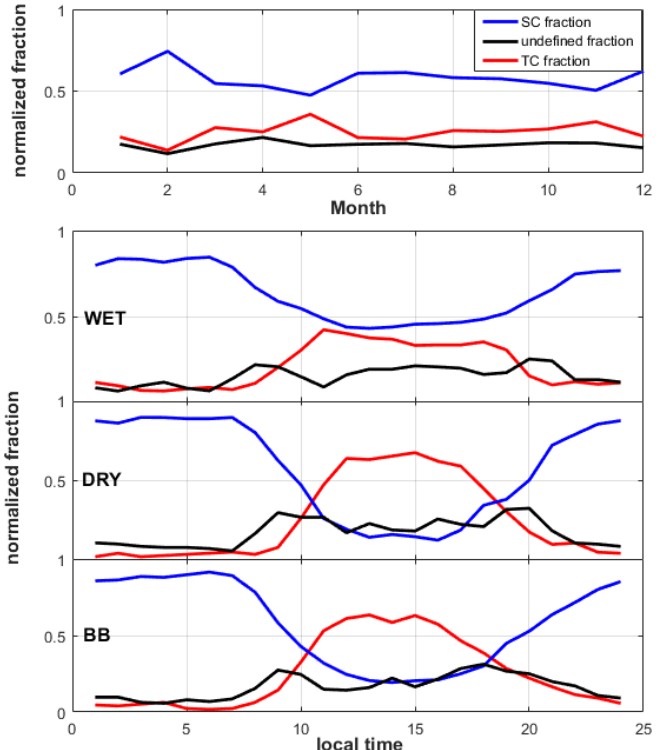

**Figure 3: Monthly and diurnal variations of turbulent conditions (TC, red), stable conditions (SC, blue) and undefined (black) fractions for each season.**

### 2.3. Identification of air mass origins at regional and meso-scales

HYSPLIT (Hybrid Single Lagrangian Integrate Trajectory, Stein et al., 2015) back-trajectories (BTs) are used in this study to investigate the aerosol particle transport to the CHC station and their properties as a function of the air mass origins. Hence, 96 hours air mass BTs are calculated every hour from the CHC station during the four years measurement period (from 2012 to 2015). WRFd04 dataset has been used to generate BTs every hour, starting at nine locations around the Chacaltaya station (within a square of 2x2 km around the station). This dataset presents the best topographic resolution for this region with spatial resolution of 1.06x1.06 km, and 28 pressure levels.

The BTs have been grouped into different clusters defined by their similarities in time and space. The Cluster Analysis method used in this study is described in Borge et al. (2007) and based on the Euclidean geographical coordinates distance and given time intervals. Figure (4a) shows the trajectory frequency plot. The opacity of each pixel is proportional to the number of BTs passing through each grid cell. Clusters are defined by using a two-stage technique (based on the non-hierarchical K-means algorithm). Six clusters have been found around the Chacaltaya station. Hence, a fraction of each cluster is assigned to each BT, and is calculated according the residence time in each cluster and their distance from the reference location (the Chacaltaya station). In order to obtain aerosol optical properties of each cluster, only a part of the back-trajectories have been selected. One BT is selected if its contribution to one cluster is high enough. For each cluster, the first 10% of the BTs have been selected by demonstrating the highest contribution to any one cluster. This firsts 10% of BTs related to each clusters and their mean paths are shown in Appendix A1.

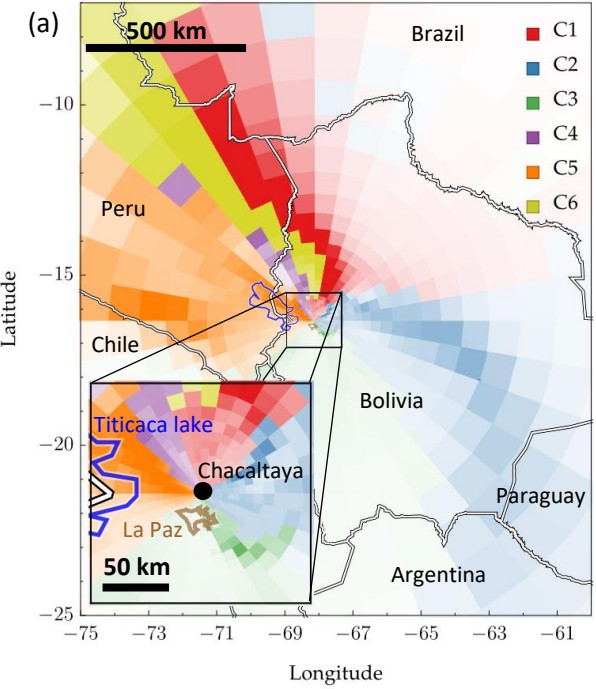

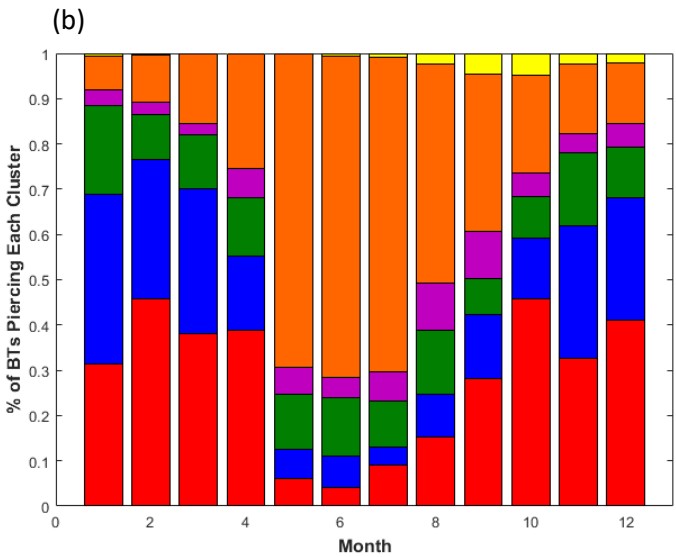

Figure 4: a. Trajectory frequency plot in La Paz region in Bolivia, with a scale of 500 km and 50 km in the lower left corner centered on CHC station. b. Monthly variation of the percentage of back-trajectories (BTs) for each cluster.

Figure (4b) shows seasonal variations of each cluster fraction. Results show that most of the air-masses influencing the CHC station come from the highlands (Altiplano), the Pacific Ocean, and along the Cordillera Real slopes in the North of the CHC station. For each cluster, a characteristic geolocation along the path of back-trajectory is identified, and acronyms are used for clarity:

- Cluster 1 (NA): Northern Amazonian Basin / North-East slope of Cordillera Real

- Cluster 2 (SA): Southern Amazonian Basin

- Cluster 3 (LP): La Paz / El Alto

- Cluster 4 (ATL): Altiplano / Titicaca lake

- Cluster 5 (APO): Altiplano / Pacific Ocean

- Cluster 6 (NES): North-East slope of Cordillera Real

Clusters 1 and 2 (NA and SA respectively) cover the entire East part of air masses, limited by the high wall formed by the Cordillera Real. These two clusters could be influenced by Amazonian Basin activities, such as BB that is extremely active from August to September, and biogenic forest emissions. Cluster 3 (LP) seems to be the main cluster representing local urban emissions, for example vehicle emissions, industrial activities, and domestic

heating. Clusters 4 and 5 (ATL and APO) can both give information of Altiplano sources (dust, urban emissions, …) but also humid air masses from Pacific Ocean and the Titicaca lake. Finally, cluster 6 (NES) has properties close to cluster 1 but with less influence from the Amazonian Basin and close to cluster 4 but with aerosol sources further from CHC station (> 100 km). All these cluster definitions will be discussed in this paper in the context of the associated aerosol particle optical properties.

Figure (4b) shows the seasonal influence of the different clusters to CHC measurements. During the dry period, air masses measured at the CHC station are mainly influenced by North-West (cluster APO), according for more than 60% of the BTs between June and July. During the wet period, the main influence is from the East (clusters NA and SA) with more than 60% of the BTs between December and April. Finally, LP, ATL and NES shares about 10% of the BTs throughout the year but with local maximum in August, September and October respectively.

## 3. Aerosol particle optical properties

### 3.2. Seasonal and diurnal variations

Monthly median scattering coefficient ($\sigma_{scat}$), absorption coefficient ($\sigma_{abs}$), extinction coefficient and single scattering albedo (SSA) from 2012 to 2015 are shown in Fig. (5a) with 25th and 75th percentiles. They are all interpolated at 500 nm using scattering and absorption Angström coefficients (Equ. (7) and (8)). Extinction coefficient and SSA were calculated from Equ. (5) and (6) respectively at 500 nm. Figure (5b) shows monthly Angström exponent values from the same dataset.

The annual median [25th percentile – 75th percentile] absorption coefficient at CHC is 0.74 Mm⁻¹ [0.43 – 1.25] at 500 nm. A clear seasonal variation can be observed with low values during the wet season (0.57 Mm⁻¹ [0.32 – 1.05] between December to March) and higher values during the dry season (0.80 Mm⁻¹ [0.52 – 1.24] between May and July). The highest values are observed from July to November (including the August-September BB period) with a median absorption coefficient of 1.00 Mm⁻¹ [0.64 – 1.70]. Similar seasonal variations are observed for the scattering coefficient, with a more pronounced increase occurring during the BB period.

(a)
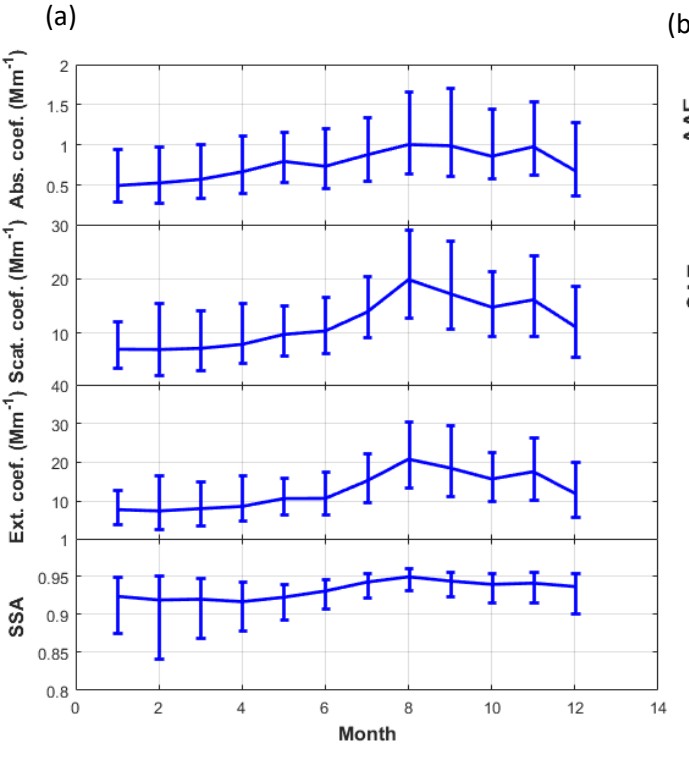

(b)
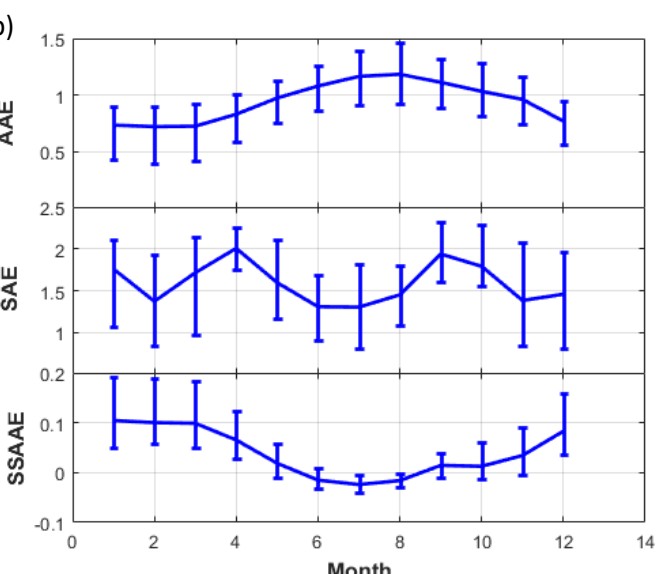

**Figure 5: Monthly (a) absorption coefficient, scattering coefficient, extinction coefficient and single scattering albedo (all at 500 nm) measured at CHC stations for 4 years, and (b) absorption Angström exponent (AAE), scattering Angström exponent (SAE) and single scattering Angström exponent (SSAAE). Solid line corresponds to the median and error bars indicate the range between the 25th and 75th percentiles.**

The median scattering coefficient of the entire dataset is 12.14 Mm$^{-1}$ [6.55 – 20.17]. Scattering coefficients are lower during the wet season (7.94 Mm$^{-1}$ [3.45 – 15.00]) than during the dry season (11.23 Mm$^{-1}$ [6.94 – 17.60]), and reach a maximum median scattering coefficient of 18.57 Mm$^{-1}$ [11.63 – 28.45] during the BB period. Regardless of the season, these values are very low in comparison to aerosol particle optical properties at lower lying stations, as shown by Chand et al. (2006) during the Large Scale Biosphere-Atmosphere Experiment in Amazonia – Smoke, Aerosols, Clouds, Rainfall and Climate (LBA-SMOCC) campaign at the Fazenda Nossa Senhora Aparecida (FNA) station (10.76°S, 62.32°W, 315 m.s.l.). Their works show that scattering coefficients and absorption coefficients reach 1435 Mm$^{-1}$ and 70 Mm$^{-1}$ respectively during important BB periods while remain at 5 Mm$^{-1}$ and 1 Mm$^{-1}$ during clean conditions.

The median extinction coefficient at CHC is 12.96 Mm$^{-1}$ [7.07 – 21.62]. This extinction coefficient range are at least one order of magnitude lower than other measurements reported during the BB period in the Amazonian Basin. This is likely due to the altitude at which the CHC station is located and, but mainly, its distance from the BB sources.

We measured a small seasonal variation of the SSA, with a median value of 0.93 [0.87 – 0.95] during wet season, 0.93 [0.91 – 0.95] during dry season and 0.95 [0.93 – 0.96] during the BB period. These observations are again different from results reported from measurements performed closer to BB sources in the Amazonian region, with SSA being higher at CHC than at the source regions. Reid et al. (1998) show that at Cuiaba, Porto Velho and Maraba, the SSA was around 0.80 from aircraft measurements during BB episodes. However, the authors report that the SSA values increase rapidly with time, i.e. from 0,85 to 0,90 in 1 or 2 days in this region (Reid et al., 1998 ; Reid et al., 2005). The remote location of the Chacaltaya station thus explain high SSA values observed in the present study.

Figure (5b) shows the monthly variations of the Angström exponents. Even, AAE values are slightly lower than expected (between 1 and 2 according Russel et al., 2010), variations of AAE and SSAAE values exhibited typical seasonal variation. Lowest values of AAE are retrieved between December and March (mean AAE value of 0.8) and highest SSAAE values are retrieved during the same period (around 0.1). A seasonal variation of these intensive optical parameters shows that different sources of aerosol influence the CHC in different season. While AAE and SSAAE values show a significant seasonal variability, SAE values are more fluctuating. The highest SAE values are observed in April and September (up to 2) and persisting low values are seen between June and August. Ealo et al., (2016) used Angström coefficients to address the nature of aerosols. Applying their analysis technique, the seasonal evolution of AAE, SSAAE and SAE can be interpreted that urban emissions (low AAE values and high SSAAE and SAE values) contributes in the wet period in the La Paz region (from December to March) whereas dust particles mostly contribute in the dry and biomass-burning period (from April to November).

Figure (6a) shows the diurnal variations of aerosol particle optical properties averaged over the wet and dry seasons and BB period, and the diurnal variation of the standard deviation of the wind direction. For extensive optical parameters, a clear increase is observed starting around 08:00 in local time. This time evolution is observed for all seasons, as the result of the diurnal variation of the turbulent layer height described in Sect. 2.3. Optically scattering and absorbing particles emitted at ground level are mixed into the turbulent layer and reach the CHC altitude due to dynamic and convective effects of the atmosphere during daytime. Indeed, the variation of the atmospheric dynamics can be observed through the variation of the wind direction with significantly stronger turbulences between 08:00 and 12:00 and during all seasons.

From the diurnal variations one can also observe that not only the daytime optical properties exhibit pronounced seasonal variation, but also nighttime coefficients do, being influenced by the SC (Fig. 3). This confirms that emissions in the region have a clear influence on both TC and SC layers which can be measured at high altitude stations continuously.

The diurnal variation of the SSA shows a clear decrease at around 11:00, when only TC particles are sampled at the station, indicating that TC particles are relatively more absorbing compared to SC particles. This observation can be explained by the local BC emission from traffic (Wiedensohler et al., 2018) and aged BB particles. Values can reach 0.90 a few hours after exhaust according Reid et al. (2005). Figure (A2) allows to identify these urban influences of the in-situ measurements at CHC station through the difference of the AAE between workdays and Sundays.

Figure (6b) also shows hourly variations of the Angtsröm exponents for the three periods. A diurnal variation is observed mostly for SSAE values for the wet period, with an increase of more than 50% of SSAAE values during daytime compared to nighttime. As from extensive optical properties, these observations can be explained by the arrival of the TC at CHC stations, with more local urban particles reaching the mountain station around 11:00.

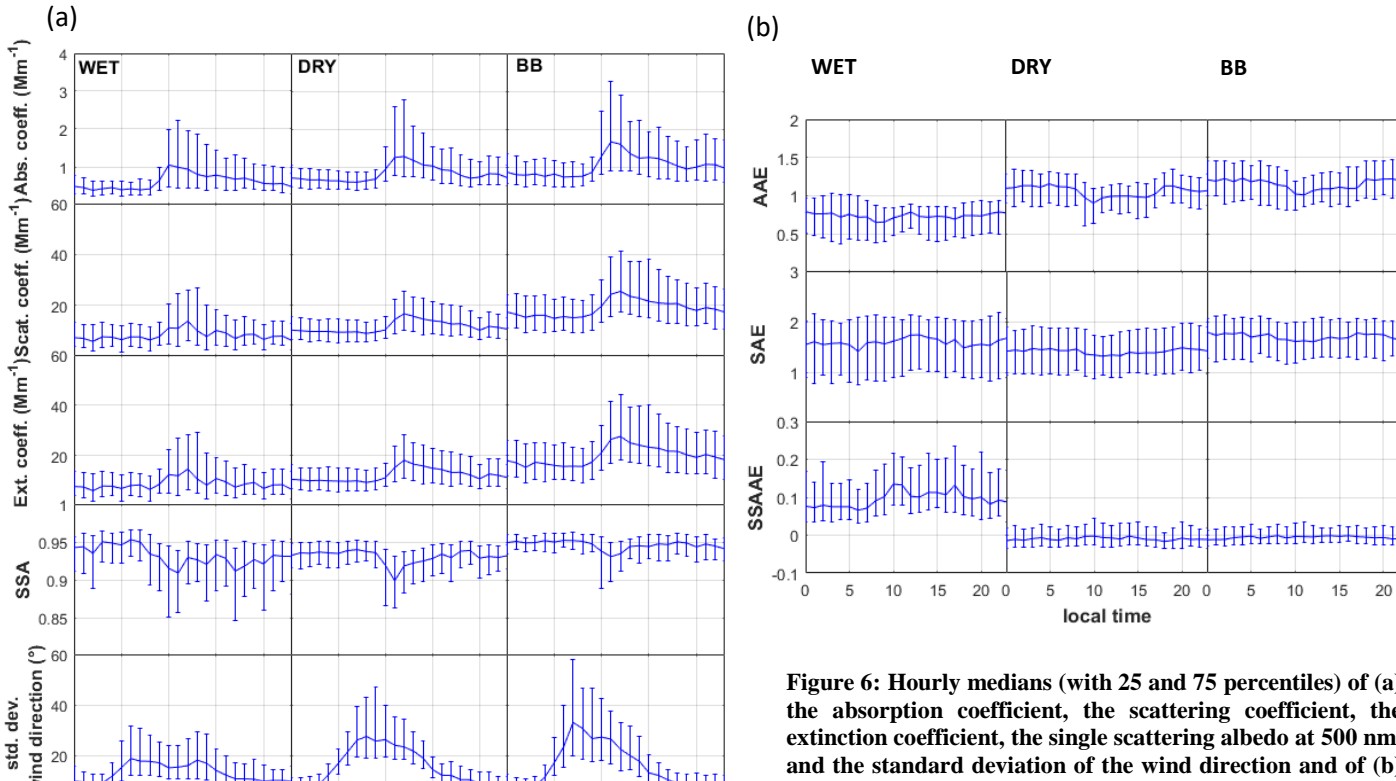

**Figure 6: Hourly medians (with 25 and 75 percentiles) of (a) the absorption coefficient, the scattering coefficient, the extinction coefficient, the single scattering albedo at 500 nm, and the standard deviation of the wind direction and of (b) the AAE, SAE and SSAAE measured at the CHC station for each season.**

### 3.3. Aerosol particle optical properties in stable and turbulent layer conditions

Using the method explained in Sect. 2.3, it is possible to characterize SC and TC optical properties separately. Median optical properties for each atmospheric condition (TC and SC) are presented Fig. (7) for each season (Wet, Dry and BB).

The optical properties of the particles sampled in the TC are different from the ones sampled in the SC, with 1.49 times higher scattering coefficients, 1.94 times higher absorption coefficients, and 1.55 times higher extinction coefficients. We observe that the difference between TC and SC is highest during the wet season and lowest during the dry season. Indeed, the mean TC to SC ratio of extinction coefficients is 1.71 during the wet season, while it is only 1.49 during the dry season, and 1.44 during the BB period. These lower TC to SC ratios indicate that the SC particles are more influenced by TC intrusions during the dry season and the BB period than during the wet season.

These results also show that emissions from the Amazonian basin have important influence on the whole atmospheric column and at the regional scale. Indeed, both SC and TC present higher extinction coefficients during the BB period than the dry season (around 2 times higher). Same observations can be made for scattering and absorption coefficients.

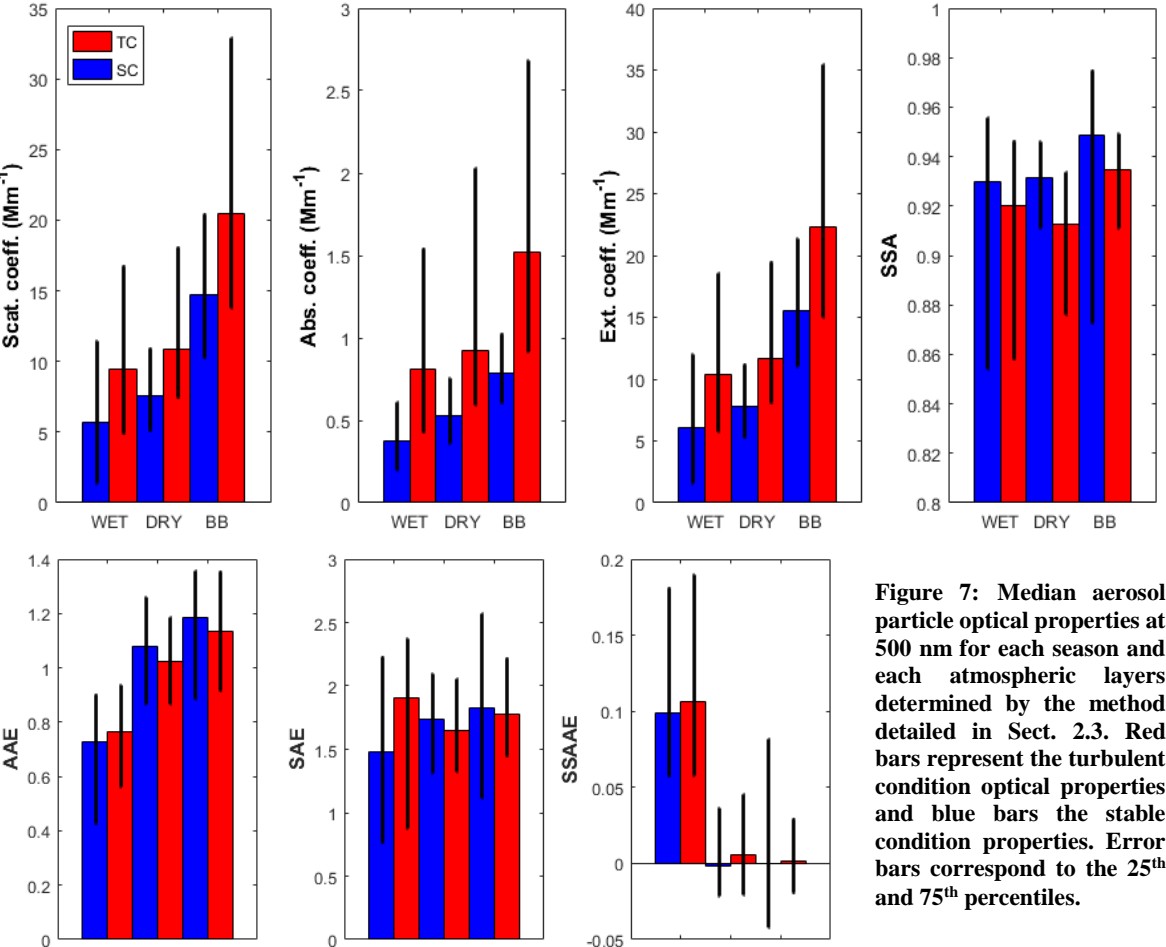

Figure 7: Median aerosol particle optical properties at 500 nm for each season and each atmospheric layers determined by the method detailed in Sect. 2.3. Red bars represent the turbulent condition optical properties and blue bars the stable condition properties. Error bars correspond to the 25th and 75th percentiles.

The SSA values do not show a strong contrast between the SC and the TC although lower values are systematically observed in the TC compared to the SC (0.93, 0.93 and 0.95 during wet, dry and BB seasons respectively in the SC, and 0.92, 0.91 and 0.93 in the TC). As discussed by Reid et al. (2005), SC aerosol particles are aged longer and transported farther than TC particles due to less scavenging effects. The long transport modifies their optical properties to slightly increase the SSA. However, the small SSA difference between TC and SC indicates that the nature of the aerosol is actually similar between the TC and SC for a given season. As shown previously, the AAE increases in the dry and BB seasons, probably due to dust transport and to BB emissions. The contribution of dust is consistent with the drastic difference in SSAAE between wet season on one side and dry and BB seasons on the other. Not only SSA values, but also AAE, SAE and SSAAE values show weak TC/SC contrast. This again illustrates that ageing processes of air masses into the full troposphere which homogenize their properties with time after emission.

### 3.4. Influences of air mass type on aerosol particle optical properties

The separation of air mass types into clusters allows us to analyze the influence of the different sources surrounding the station on their aerosol particle optical properties. The seasonal variability of aerosol particle optical properties may be attributed to a seasonal variability in the air mass types arriving at the station. As shown Fig. (4), air masses coming from the North-West (Clusters APO) dominates during the dry season, whereas air masses are from the East of the station. (Cluster NA and SA) plays a major role during the wet season. Figure (8) shows TC and SC

optical properties. For each cluster and each season, the left bar indicates the TC property, and the right bar the SC property.

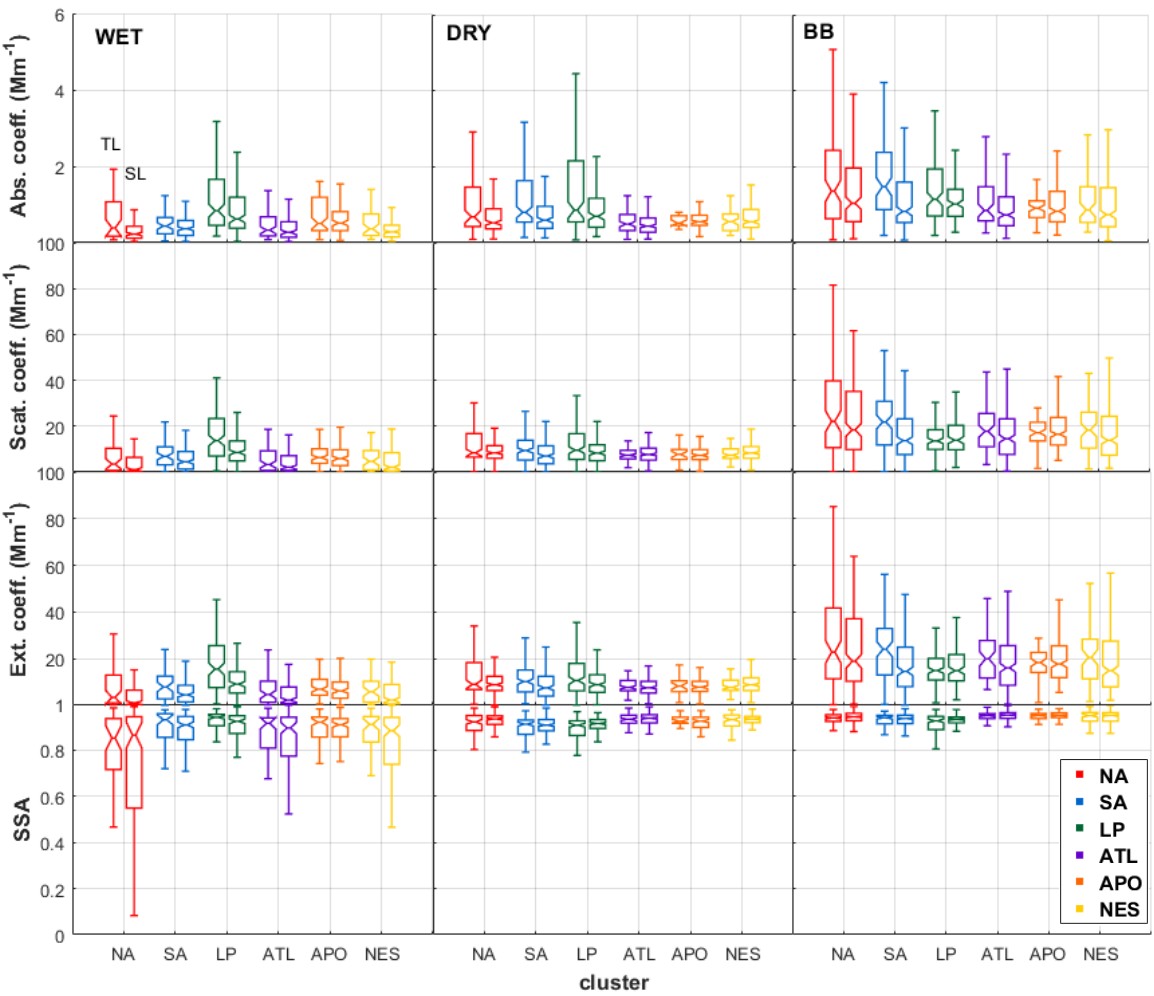

**Figure 8: Aerosol particle optical properties from Chacaltaya measurements from 2012 to 2015 at 500 nm for TC (left bar), and SC (right bar) layers for each cluster and the three periods.**

A strong air mass type dependence of the aerosol optical properties is found during the wet season. The highest extinction coefficients are found within air masses originating from the urban area of La Paz / El Alto (LP in green) with a median value of 13 Mm$^{-1}$. This value is significantly larger than other air-masses, which remains at less than 7 Mm$^{-1}$. The exceptionally high extinction coefficient can be mainly due to particle emissions from traffic in La Paz (Wiedensohler et al., 2018), despite the effect of wet deposition during this period. The lowest SSA are measured during the wet period, mainly within the NA air masses. This may be explained by important heating activities from the Yungas region, on the North-East slope of Cordillera Real at this period. Due to the wet deposition, aerosol particle life-time is significantly decreased and the main part of aerosol particle optical measurements at CHC is from low altitudes (TC). Indeed, median TC extinction coefficients are from 10% (for APO) to 200% (for NA) higher than values in the SC.

During the dry season, extensive optical properties are larger than during the wet season for all clusters except for the cluster from the urban area of La Paz / El Alto (LP). The extinction coefficients are by more than 50% larger, with median values around $10 \pm 2$ Mm$^{-1}$. The soar of the extensive optical properties is due to low wet deposition rate during the dry season that extends aerosol lifetime. The extended aerosol lifetime allows local emissions to reach the in-situ station.

A further clear increase of all extensive optical properties are observed during the BB period. Extinction coefficients increase by 42% (LP) to 203% (NA). The cluster analysis shows that BB events impact atmospheric properties regionally influencing all clusters. However, higher increases are observed for air masses coming from the East (more than 80% higher for NA, SA, LP and NES) compared to direction from the West (45% to 70% higher), directly linked to intensive anthropogenic activities in the Amazonian Basin during this period. During extreme events, average extinction coefficients at CHC measurements can reach 247 Mm$^{-1}$ (for NA). A strong influence of BB emissions appears in TC measurements with extinction values more than 160% higher than during the dry season, but a significant increase is also observed in the SC measurements (around 110% higher to dry season for North-East air masses).

In addition to the classification of air masses into different clusters, we further classify aerosol particle types from their optical properties to characterize the influence of the different sources in this region.

Figure (9) shows the correlation of the Ångström exponents for absorption (AAE), scattering (SAE) and Single Scattering Albedo (SSAAE), for each season (symbols), each cluster (colors) and each tropospheric condition (filled marker for the TC and un-filled markers for the SC).

As shown in Fig. 5, low AAE values, especially during the wet season, can be explained by important reduction of dust and less biomass burning particles due to more efficient removal. As observed previously, Fig. (9) demonstrates that the wet season (diamonds) in this region is mainly influenced by a different source from the wet and the dry season. Thus, the wet season presents positive SSAAE and AAE close or lower than 0.9, while dry season and BB period present SSAAE close to 0 and AAE higher than 0.9. A linear relationship between AAE and SSAAE values is observed and illustrates that mainly urban emissions drive aerosol particle properties during the wet period, and that mainly dust emissions drive aerosol particle properties during the dry season and the BB period. Indeed, the large covering of arid surfaces on the Altiplano (West of the CHC station) presents an important source of dust. This result also indicates that whatever the air mass type, and the atmospheric condition, ground emissions are influencing the optical properties of the whole atmospheric column and at a regional scale. In addition to dust emissions, BB period also demonstrates a significant contribution of BB combustion particles with higher median AAE values than during the dry season. Except for the NES cluster (West side of the Bolivian cordillera), AAE values are retrieved between 1.1 and 1.3 during BB period and between 0.9 and 1.1 during the dry season.

Even though there are dominant aerosol sources for each season as demonstrated in Fig. (8), the scatter plots of Ångström exponents in Fig. (9) provide additional insights into air mass origins. During the dry season, the scatterplot of AAE and SAE shows an important contribution of urban emissions in addition to the dominant dust aerosol. For some clusters, characteristics of urban emission is observed with AAE close to 1 and SAE higher than 1.4 for some clusters. During the BB period, a strong TC/SC dependence is seen for La Paz / El Alto air masses (cluster LP). The AAE value for TC indicates urban pollution effects (AAE below 1.1) whereas the AAE for SC shows an influence of BB emissions (AAE close to 1.3). A similar SC dependence can also be observed for NA and APO clusters. Because AAE values are powerful tracers to separate urban and BB influences on aerosol particle optical properties, the TC/SC dependence clearly demonstrates the influence of Amazonian biomass-burning on Chacaltaya in-situ measurements during the BB period within the SC. Because BB particles are mainly emitted from the East part of the Bolivian Cordillera, NES air masses are less influenced by these sources and present the lowest AAE values during the BB period.

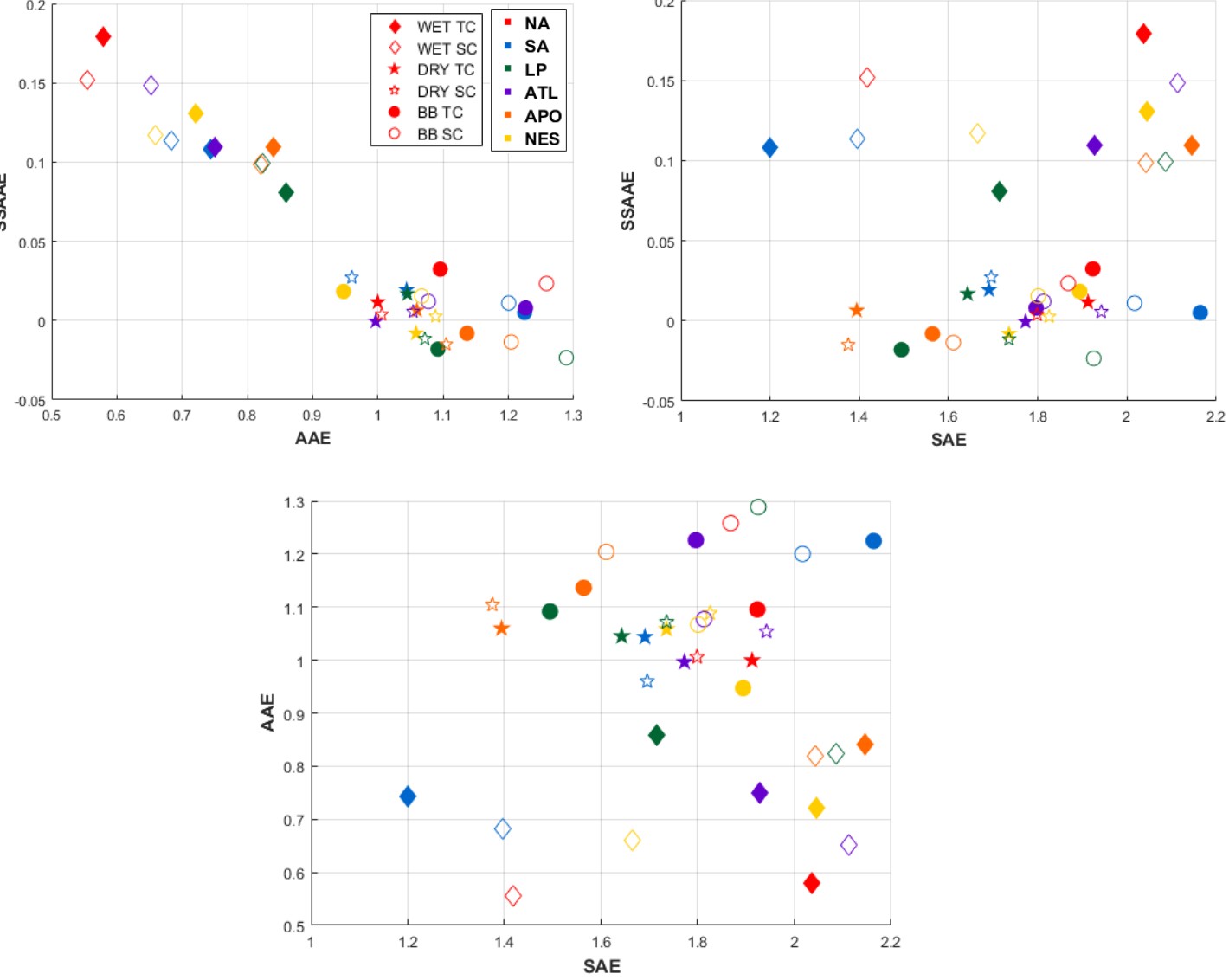

**Figure 9: Wavelength dependence of optical properties measured at Chacaltaya station for each cluster (colors) and each season (markers) as parametrized by Ångström exponents. Diamonds correspond to median values during the WET period, stars correspond to the DRY period and circles correspond to the BB period.**

The distribution of SAE is more spread, and the value depends on atmospheric conditions, clusters and seasons. In addition to urban influences, during the BB period and the wet season, LP air masses are also affected by dust particles, especially in the TC, with lower SAE values. The opposite is observed from NA air masses. While the main influence in these two air masses remains from urban emissions, it can be noticed that the lower part of the atmosphere (TC) in LP air masses are more affected by local dust particles than at the higer part of the atmosphere (SC). In NA cases, also observed for APO air masses, SC measurements are less influenced by urban particles due to longer distance between CHC station and urban emissions than in LP cases.

Table 2 summarizes the median Angström exponents measured at the CHC station for turbulent conditions (stable conditions in parenthesis). According to these values and as discussed above, aerosol types for the turbulent conditions (and stable conditions in parenthesis) are given.

| Cluster | season | SAE | AAE | SSAAE | Aerosol types |
|---------|--------|-----|-----|-------|---------------|
| NA | WET | 2,04 (1,42) | 0,58 (0,56) | 0,18 (0,15) | urban (dust/urban) |
| | DRY | 1,91 (1,80) | 1,00 (1,01) | 0,01 (0,004) | urban (dust) |
| | BB | 1,92 (1,87) | 1,10 (1,26) | 0,03 (0,02) | dust/BB (dust/BB) |
| SA | WET | 1,2 (1,40) | 0,74 (0,68) | 0,11 (0,11) | urban (urban) |
| | DRY | 1,69 (1,70) | 1,04 (0,96) | 0,02 (0,03) | dust (dust) |
| | BB | 2,16 (2,02) | 1,23 (1,20) | 0,005 (0,01) | BB (BB) |
| LP | WET | 1,71 (2,09) | 0,86 (0,82) | 0,08 (0,10) | urban (urban) |
| | DRY | 1,64 (1,74) | 1,05 (1,07) | 0,02 (-0,01) | urban (dust/urban) |
| | BB | 1,49 (1,93) | 1,09 (1,29) | -0,02 (-0,02) | dust (dust/BB) |
| ATL | WET | 1,93 (2,11) | 0,75 (0,65) | 0,11 (0,15) | urban (urban) |
| | DRY | 1,77 (1,94) | 1,00 (1,05) | -0,001 (0,006) | dust (dust/urban) |
| | BB | 1,80 (1,81) | 1,23 (1,08) | 0,008 (0,01) | dust/BB (urban) |
| APO | WET | 2,15 (2,04) | 0,84 (0,82) | 0,11 (0,10) | urban (urban) |
| | DRY | 1,39 (1,38) | 1,06 (1,10) | 0,006 (-0,02) | dust (dust) |
| | BB | 1,56 (1,61) | 1,14 (1,20) | -0,008 (-0,01) | dust/BB (dust/BB) |
| NES | WET | 2,05 (1,67) | 0,72 (0,66) | 0,13 (0,12) | urban (urban) |
| | DRY | 1,74 (1,83) | 1,06 (1,09) | -0,008 (0,003) | dust/urban (dust) |
| | BB | 1,89 (1,80) | 0,95 (1,07) | 0,002 (0,02) | dust/urban (urban) |

**Table 2: Median aerosol Angström exponents of turbulent condition (stable condition) for each cluster and seasons measured at the CHC station and resulting aerosol types.**

**Conclusions**

Chacaltaya station is currently the unique high-altitude atmospheric observatory in the Andes. The location of the station allows to sample air masses of different types (mainly urban, biomass-burning and dust particles). Measurements have been run over a long-term period at a high temporal resolution and a large set of instruments. This study reports on the impact of several aerosol sources in South-America through the variability of aerosol particle optical properties. We show that the Central Andean region (cordillera Real) is characterized by median annual values of absorption, scattering and extinction coefficients of 0.74, 12.14 and 12.96 Mm$^{-1}$ respectively. Results also show the effect of the two main seasons, a dry and a wet season, on aerosol particle optical properties characteristic of different source influences. Diurnal variations are also observed due to Atmospheric Boundary Layer dynamics influencing this high altitude location.

The topography of the surrounding region also gives unique opportunities to sample aerosol particle optical properties within different atmospheric conditions. For each season, stable conditions (SC) have been identified in contrast to turbulent conditions (TC) using the standard deviation of the wind direction. Even TC is usually attributed to mixing layer, SC can be undoubtedly attributed to free tropospheric or residual layers.

Every year, from July to November, this region is influenced by important biomass-burning activities at the regional scale. The present study clearly demonstrates the regional impacts of these activities. Results show higher scattering and absorption coefficients during the BB period (44% to 144% increase compared to the dry season) that can be observed in all tropospheric layers. The present study has hence demonstrated that BB particles are efficiently transported to the higher part of the troposphere (Stable conditions) and over long distances (more than 300 km long). However, differences in the optical properties between different air mass types are less pronounced in the SC than in the TC, which can be mainly explained by the longer life time of the aerosol particles within the higher troposphere.

One of the main aerosol sources in the Bolivian plateau is the urban area of La Paz / El Alto. It contributes significantly to optical properties of the atmosphere due to important traffic emissions and industries (Wiedensohler et al., 2018). In addition to BB activities, the urban area with 1.7 million inhabitants, located at 17 km south to the CHC station between 3200 and 4000 m a.s.l, was found to contribute significantly to the optical

characteristics of the aerosol particles sampled at CHC. The lowest single scattering albedo values (median of 0.85), attributed to incomplete combustion, was observed for back-trajectories from the urban area of La Paz / El Alto during the wet season, and the same air-mass has the highest extinction coefficient during the wet season. This strong signature of pollution aerosols is also highlighted by the wavelength dependence of the absorption Angström exponent (AAE) both in TC and SC.

Finally, the arid plateau of the region has also demonstrated regional impact. In addition to urban and BB influences, the wavelength dependence of the single scattering albedo (SSAAE) measured at CHC highlights a main dust influence during the entire dry season with SSAAE values close to 0. This influence is no longer observed during the wet season due to particle scavenging and less dust uprising due to wet soils.

| Aerosol type | SAE | AAE | SSAAE |
|---|---|---|---|
| Dust | - | > 0,9 | [-0,05 ; 0,05] |
| Urban pollution | > 1,4 | < 0,9 | > 0,05 |
| Biomass burning | - | > 1,1 | [-0,05 ; 0,05] |

**Table 3: Updated Angström exponent values expected for aerosol types at the CHC station.**

A new Angtsröm exponent classification can then be defined for measurement at the CHC station and is reported Table 3. Thresholds are close to the ones proposed by previous works (Dubovik et al., 2002 ; Collaud Coen et al., 2004 ; Clarke et al., 2007 ; Russel et al., 2010) but adapted to CHC's instruments and particular atmospheric conditions.

The in-situ measurements of the high-altitude station of Chacaltaya provide useful information on the different aerosol sources in this region. Thus, they can be used to validate satellite products as Cloud Aerosol Lidar and Infrared Pathfinder Satellite Observations (CALIPSO) LIDAR measurements of the vertical aerosol profiles, when chosen at the adequate time of the day. We also found that most aerosol intrinsic properties were very similar over the whole atmospheric column, thus indicating that those can also be used to validate Moderate Resolution Imaging Spectroradiometer (MODIS) measurements of columnar aerosol particle optical depth over the bright region at high elevation.

**Author contributions:** P.G., I.M., and F.V., with the help of the UMSA, carried out the measurements at the station. M.A., K.S., A.W. and P.L. supervised the project. D.A. and F.V. did trajectory analyses and I.M. developed the method to discriminate stable and turbulent conditions. A.C wrote the manuscript with the help of D.A. R.K, G.M, and T.M., brought instrumental instructions to convert and correct measurements. N. M., M.P. and K.W. helped shape the research and analysis. All authors discussed the results and contributed to the final manuscript.

**Acknowledgements**. Day-to-day operations at CHC station are under the responsibility and support of UMSA through the Institute for Physics Research (Laboratorio de fisica de la Atmosfera).
This work was also accomplished in the frame of the project ACTRIS-2 (Aerosols, Clouds, and Trace gases Research InfraStructure) under the European Union – Research Infrastructure Action in the frame of the H2020 program for "Integrating and opening existing national and regional research infrastructures of European interest" under Grant Agreement N°654109.
We acknowledge the support from IRD (Institut de Recherche pour le Développement) under Jeune Equipe program CHARME awarded to LFA, by Labex OSUG@2020 (Investissements d'avenir – ANR10 LABX56) and INSU-CNRS under the Service National d'observation programme CLAP and ACTRIS-FR.
We gratefully acknowledge Souichiro Hioki for his help on english corrections and proofreadings.

**Appendix:**

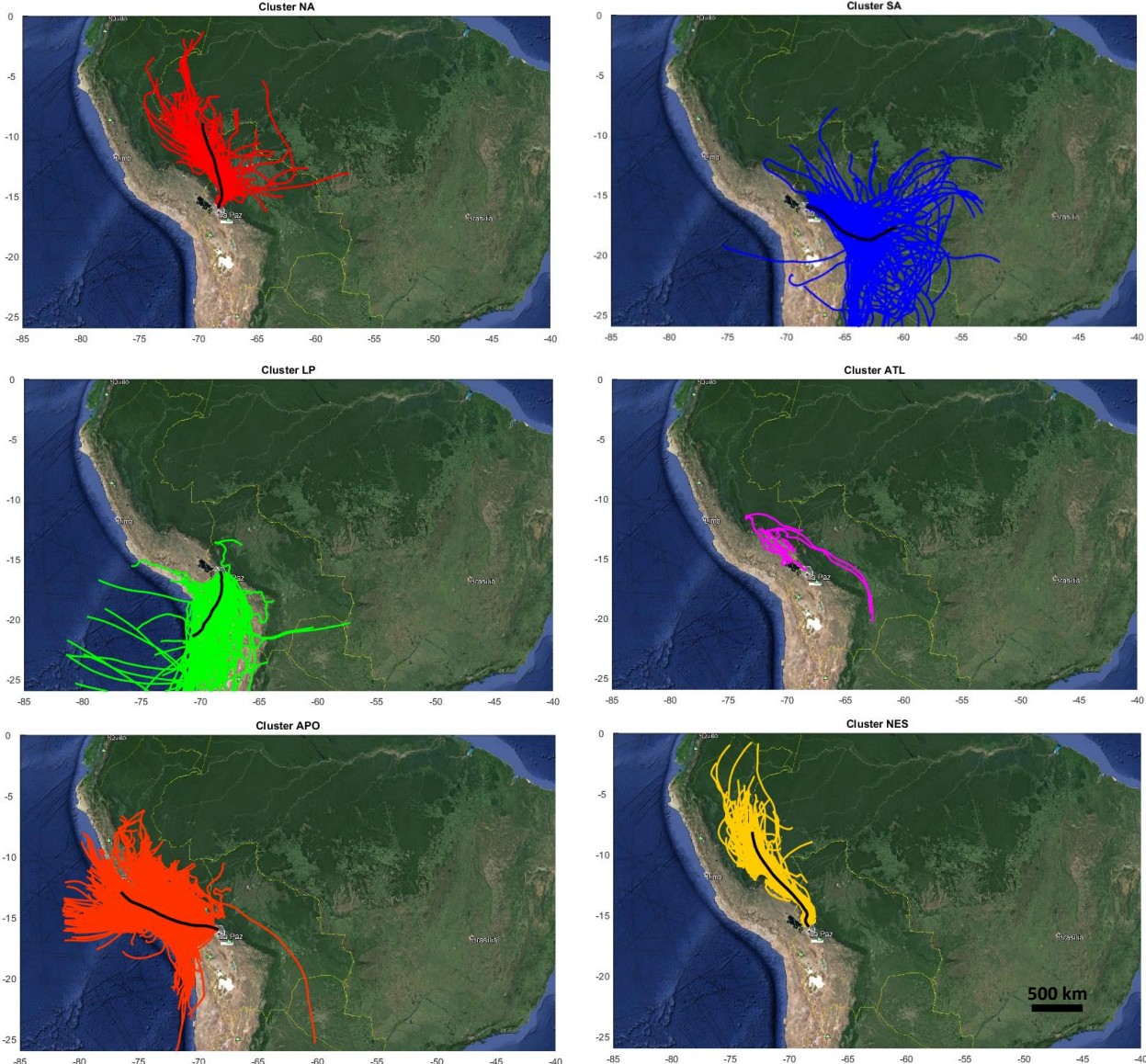

**Figure A1: Selected 96-hours back-trajectories for the six clusters obtained from the Borge et al. (2007) method. The black line corresponds to the main back-trajectory.**

For each hour of the period of the study, nine back-trajectories have been used to describe the mean influence at
Chacaltaya station. The nine BTs start within a square of 2 km by 2 km around the station. The mean BT has been calculated from these nine BTs and generated every hour from January 2012 to December 2015. Clusters are defined according the Borge et al. (2017) methods using a two-stage technique (based on the non-hierarchical K-means algorithm). The Borge et al. (2007) method allows to attribute to each mean BT a fraction of each cluster according to their time residence into the cluster and their distance from the CHC station. Hence, BTs are sorted
according to their representativeness in each cluster. The first 10% of them are used in the present study and are reported in Fig. (A1).

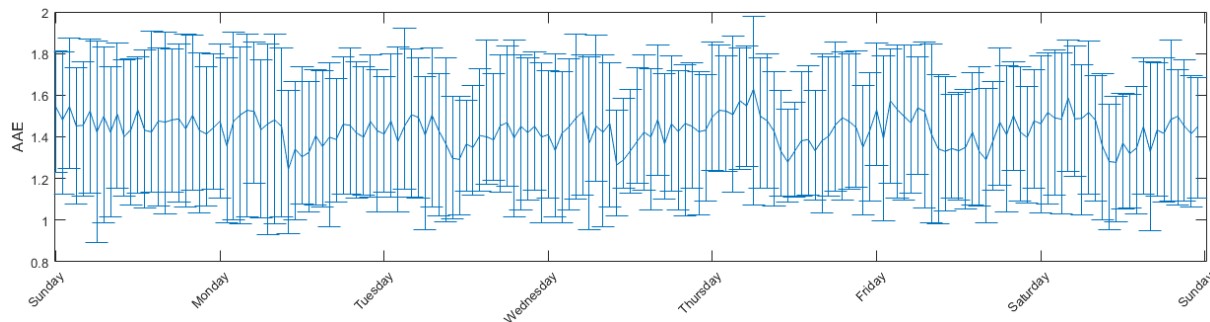

**Figure A2: Weekly variation of the Aerosol Angström Exponent (AAE) for the whole dataset from 2012 to 2015. The medians and their 25th and 75th percentiles from Sundays to Saturdays are repsresented.**

The weekly variation of the Absorption Angström Exponent (AAE) is shown in Fig. (A2) for the four years' dataset. This representation of in-situ measurements at Chacaltaya station allows a better discrimination of anthropogenic influence on the aerosol optical properties.

Net decrease of the AAE is observed for every working day at about 10:00 - with median values of 1.2 in contrast to around 1.5 in the beginning and the end of the day – whereas Sundays clearly show constant (±0.05) values of AAE for the all day. These observations show that aerosol concentrations measured on Chacaltaya greatly depends on the activities in the urbanized area below the station.

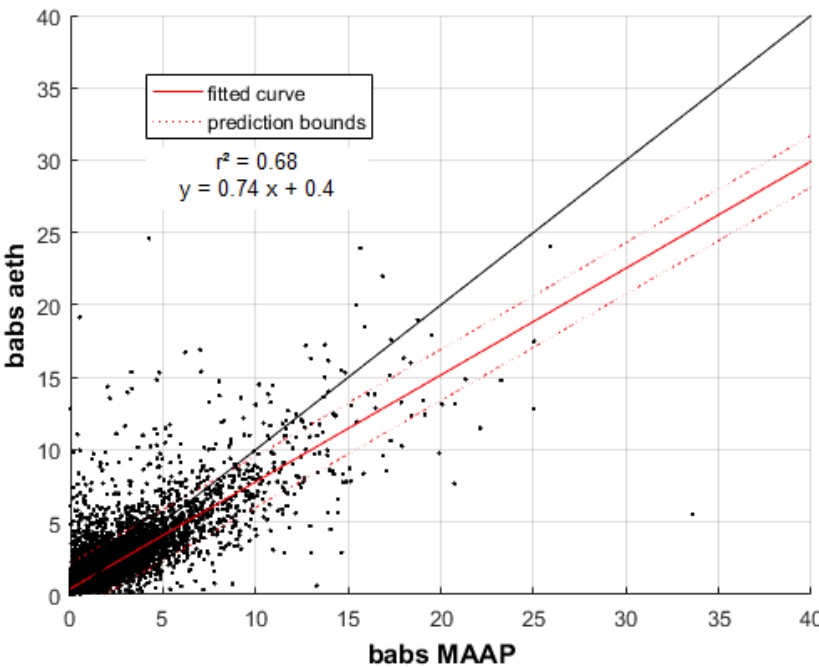

**Figure A3: Comparison of absorption coefficients measured at 635 nm according aethalometer and MAAP measurements from the CHC dataset between 2012 and 2014. Black line corresponds to the 1 to 1 fit.**

Figure (A3) shows comparison of absorption coefficients at 635 nm measured by the aethalometer and the MAAP
at CHC station from 2012 to 2014. Because MAAP measurements can measure the aerosol particle absorption coefficient with a better accuracy (Saturno et al., 2017), this study validates the correction method by Weingartner et al. (2003) that is applied to the aethalometer measurements as preconized by ACTRIS (Müller et al., 2011a ; Drinovec et al., 2015).

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
