# Peer review of "Biomass-burning and urban emission impacts in the Andes Cordillera region based on in-situ measurements from the Chacaltaya observatory, Bolivia (5240 m a.s.l.)"

_Atmospheric Chemistry and Physics, 2019_

## Referee Comment (RC1) · Anonymous Referee #2 · 11 Aug 2019

This work reviews the optical properties of aerosol sampled in a mainly free tropospheric site in Bolivia for a period that spans 4 years. The authors have performed a very comprehensive and thorough analysis of their results categorizing the optical properties of aerosol in the area based on the layer sampled (FT or PBL), based on source region and on seasons. The manuscript provides a rather complete picture of the aerosol optical properties of Chacaltaya with the only information missing is the composition of the measured particles. Even though, it is understood that such information (on composition) cannot be included in this work, a short summary would be

more than welcome. I recommend that this work is published with only some minor additions which I list below

Please add a table and summarize in a small paragraph what type of particles (dust, urban, ..etc) are expected to be sampled in each season, layer and source region based on the types of categorization performed in this work. This information is available, but scattered throughout the manuscript and if you compile into one small paragraph the reader will be greatly assisted in understanding your work.

The source region analysis performed in this work is puzzling. I am not sure how source regions have been distinguished. As an example C6 and C4 seem to overlap on Fig 4b. The same holds for source regions C1 and C2. There is a second graph in the lower left corner of Fig 4b for which I could not find any explanation. What is this graph about and how it is different than the main one of Fig4b? Please improve the caption of Fig 4 to include all information so that the reader can decipher the plots easily. Some of the info required to do so are found in the text but definitely the info provided is not enough. Since you have performed this analysis for 4 years, the individual trajectories for each source region should be shown in a separate (for each source region) graph in the Appendix. Please also add another plot showing the average trajectories for each source region on a map. Hysplit has this ability to produce average trajectories and so other software that are free to use. Personally I would recommend that the average trajectories graph for each cluster to be included in Fig 4. However it is not mandatory.

There is a problem with the term $\varepsilon$ in Eq 11. Is sa and ca are the sine and cosine of the same angle then by definition $\varepsilon=0$ regardless. This is due to the well known formula of $\cos 2\theta + \sin 2\theta = 1$. I suspect a typo. In addition if $\sigma\theta$ corresponds to the 15 minute average wind direction as stated in Line 179 what is the $\theta(15)$, $\theta(30)$, $\theta(45)$, $\theta(60)$ of Eq. 10. I thought they denoted different time intervals. Please spend some effort to explain further how the classification shown in Fig.2 is performed. In other words explain further what is discussed in Lines 181 and 182.

Despite that most of this work relates to phenomenology, there are two important findings. These are the very low AAE reported during the wet season and the linear relationship between SSAAE and AAE observed during the wet and dry seasons. I am wondering if such low AAE have been reported elsewhere in literature. Please discuss. Can the authors provide an explanation on the linear relationship observed in Fig9a?

There is a typo in the caption of Appendix Fig A1. Aerosol should probably be absorption and for the entire dataset instead of the all dataset

---

## Referee Comment (RC2) · Anonymous Referee #4 · 3 Sep 2019

This paper provides an interesting overview of the aerosol particle properties observed at the high mountain station Chacaltaya in the South America (Cordillera Real). The topic is of interest for ACP and the paper is generally well written. The scientific approach is sufficiently robust and the presentation of data and results is fair. Nevertheless, some points should be better addressed before publications. In particular, the authors should better discuss the caveats related with the back-trajectories analysis as well as provide more details and information about the experimental methodologies (e.g., no information about data generation, uncertainty characterization are provided).

[Figure]

In the following you can find my specific comments. ———————————————
————————————————————————————-

Abstract. In the last sentence, the authors claimed that "CHC provides first evidences of impact of emission from Amazonian basin far away from their source". Be more specific. Which "far away" means? Please, in the site description provide distance of CHC from Amazonian basin

Line 71:please provide wavelengths

Line 92 - 105: this section ìs hard to follow. I would recommend to add a table with the different threshold values for each type of particles (dust, pollution, biomass burning) for the different Angstrom exponents (AAE, SAE, SSAE)

Line 103:please correct "bellow"

Line 135: please clearly state which kind of compensation must be applied to aethalometer data

Line 140. was the mass coefficient provided by the manufacturer independently assessed and validated by others? If yes, provide references, if not, provide adequate comments

The method for deriving the absorption coefficient it is not clear. Equation 2, what is C.R(ïĄň)?. Equation 3 it is also not clear: please describe the contribution of each member/factor. What ln(10%) and ln(50%) represent? Why the factor R should be adjusted? What do you mean for "spot" change? Why the absorption coefficient should be the same before and after the spot change?

Equation 4: what "mEBC" is? Does QEBC is equal to QBC reported by line 148?

Line 155: the authors stated that the aethalometer measurement at 635 mm is unstable. Quantitatively, what does this mean? Do you are able to provide threshold value that other users can apply to evaluate if their own measurements are unstable? In general, which QA/QC framework/procedures did you apply to all the suite of measurement discussed in this work? Please describe air inlet system and calibration strategies for the considered instrumentations (i.e. nephelometer, aethalometr, MAAP). Please quantify uncertainties related to each of these measurements.

Also equation 10 is not clear: what $\sigma\ddot{I}\acute{t}2$ (x) with X=12,30,45,60 represent? I think that the vocabulary used by the authors can be misleading. More than layers this methodology can be able to discriminate turbulent versus stable (or more stable) conditions at the measurement site. Please change nomenclature.

Line 189: why was the residual layer excluded by the analysis? Does this mean that the residual layer conditions are embedded in what the author defined as "stable" layers? Please, better specify this point since this can have implications for the interpretation of results.

Line 193: I think that "morning" must be changed by "night"

Line 204: what BT set is used for the cluster analysis (12 hours or 96 hours)? Why different TRJ lengths were considered/calculated? Is the trajectory calculation set-up changing for the 12 and the 96 hours BTs? The authors did not provide any indication about the meteorological files (which are? Which horizontal and vertical resolution?) used for BT calculation nor about calculation set-up (which starting heights? single or multiple starting points around the station locations). The resolution of the input metro files is particularly important in this mountain region, I guess. Please comment on that and provide caveats about the effective reliability of trajectories in this region. This point is critical for interpretation of results.

Figure 4a is hard to understand and the comparison among the different cluster is challenging. Maybe, it can help to use a stack bar plot with 1 bar for each single month composed by the contribution from each different single cluster.

Also Figure 4b is difficult. The geographical boundaries are not clear at all. The same

is true for the topographic features. Most of the locations listed in the legend are meaningfulness for readers not used with the region (are these villages, cities, regions?). For these reasons, it should be strongly improved.

Line 212: sentence starting with "Thus, for each cluster,..." isnt.t clear: what do you mean for "events"? " When the cluster have the most influence": what does it mean?

Line 263: Since the extinction is the sum of absorption and scattering, and scattering » absorption, the similarity between extinction and scattering is trivial.

Line 377 - 376: please better explain. in which way the AAE values are impacted by the aethalometer variability. Do you mean that the uncertainty of aethalometer is enhanced during wet season? For which reason? How this impact results robustness (please discuss in the conclusions)?

Line 401: the decrease of urban particle influence within air-masses from La Paz during "turbulent" conditions (in which I expect more efficient transport from the lower layers to CHC) is rather surprising. Does this indicate some inaccuracies in the local TRJ calculation or in the turbulent conditions identification?

Line 421:I agree that the transport to higher troposphere layers was supported but the spread over long-range (please, quantitatively specify what do you mean for long-range) is more a (reliable) hypothesis.

Line 435: the authors concluded that an effect of dust is visible during the entre dry season. However, looking at figure 9, SSAAE is mostly >0 during the dry season which contradict this (see also line 103). Please comment and/or rephrase.

---

## Referee Comment (RC3) · Anonymous Referee #3 · 10 Sep 2019

The paper by Aurelien et al. presents a detailed analysis of aerosol optical properties at a remote site of Andean mountains. The measurements are long-term and, therefore, certainly credible in terms of seasonal cycles, however, the novelty is mainly based on somewhat underexplored and sensitive region and that alone does not constitute scientific novelty. The authors make up for publishable results by careful and thorough data analysis separating dataset to represent stable and turbulent atmospheric layers alongside detailed trajectory analysis. The paper can be accepted for publication after addressing mainly minor comments. Last but not least English can be improved with

the help of senior co-authors.

Conclusions could be more concise if a summarising diagram with the main transport patterns and corridors were presented. What matters is not a repeat of study results, but emphasising something about lasting regional impacts. Otherwise, it is just another study of optical properties at a different location.

Minor comments

Line 31. Resulting in lower atmospheric...

Line 33. different aerosol sources.

Line 35. on average, instead of "in average".

Line 41. ...increase in the extinction...

Line 44. How far away? Is stable layer normally the upper layer above the boundary layer or is it free troposphere? I understand it is based on statistical treatment, but the abstract should convey the message without reading all the details.

Line 106-113. There have to be goals, not summary and justification of what and how the study has done.

Figure 1. In insert would help to visualise in the larger region, especially tha the Figure covers only appr. 50x50km.

Line 125. The papers deals with impacts over much larger region, therefore, it is important to describe that larger region, e.g. extending to 200km.

Line 193. Use past tense as measurements represent the past not present day.

Line 276. Correlation is a scientific term, therefore, cannot "correlate to seasons". Use "...values exhibited typical seasonal variation".

Line 327. I am not sure I follow why SL particles are aged longer and transported farther. Please elaborate.

---

## Author Comment (AC1) · 22 Oct 2019

Authors would like to thank the reviewer for his/her interest in the work and the constructive suggestions. Corrections allow us to clarify the method used for clustering and highlight the main goals of the paper.

The atmospheric stability is used as a tracer to differentiate the atmospheric layers. As recommended, Stable Layer (SC) and Turbulent Layer (TL) are have been replaced by Stable Conditions (SC) and Turbulent Conditions (TC). This was also requested by other 2 reviewers.

In the following, we provide answers to the reviewer's comments and list the modifications made in the manuscript.

This work reviews the optical properties of aerosol sampled in a mainly free tropospheric site in Bolivia for a period that spans 4 years. The authors have performed a very comprehensive and thorough analysis of their results categorizing the optical properties of aerosol in the area based on the layer sampled (FT or PBL), based on source region and on seasons. The manuscript provides a rather complete picture of the aerosol optical properties of Chacaltaya with the only information missing is the composition of the measured particles. Even though, it is understood that such information (on composition) cannot be included in this work, a short summary would be more than welcome. I recommend that this work is published with only some minor additions which I list below.

Please add a table and summarize in a small paragraph what type of particles (dust, urban, ..etc) are expected to be sampled in each season, layer and source region based on the types of categorization performed in this work. This information is available, but scattered throughout the manuscript and if you compile into one small paragraph the reader will be greatly assisted in understanding your work.

**Answer:** Three tables have been added to the text which summarize information from the text. The Table 1 summarizes ranges of Angström exponent for the different aerosol types. Table 2 details the median values of the Angström exponent for each cluster, season and atmospheric stability. Table 3, on the conclusion, suggests a new Angtsröm exponent definition for the different aerosol types.

**Modifications:**

| Aerosol type | SAE | AAE | SSAAE |
|---|---|---|---|
| Dust | Close to 1 | Close to 1 | Below 0 |
| Urban pollution | Close to 2 | Close to 1 | Higher than 0 |
| Biomass burning | | Close to 2,1 | |

Table 1: Expected aerosol type and their optical properties for each cluster according season and atmospheric stability.

| Cluster | season | SAE | AAE | SSAAE | Aerosol types |
|---|---|---|---|---|---|
| NA | WET | 2,04 (1,42) | 0,58 (0,56) | 0,18 (0,15) | urban (dust/urban) |
| | DRY | 1,91 (1,80) | 1,00 (1,01) | 0,01 (0,004) | urban (dust) |
| | BB | 1,92 (1,87) | 1,10 (1,26) | 0,03 (0,02) | dust/BB (dust/BB) |
| SA | WET | 1,2 (1,40) | 0,74 (0,68) | 0,11 (0,11) | urban (urban) |
| | DRY | 1,69 (1,70) | 1,04 (0,96) | 0,02 (0,03) | dust (dust) |
| | BB | 2,16 (2,02) | 1,23 (1,20) | 0,005 (0,01) | BB (BB) |
| LP | WET | 1,71 (2,09) | 0,86 (0,82) | 0,08 (0,10) | urban (urban) |
| | DRY | 1,64 (1,74) | 1,05 (1,07) | 0,02 (-0,01) | urban (dust/urban) |
| | BB | 1,49 (1,93) | 1,09 (1,29) | -0,02 (-0,02) | dust (dust/BB) |
| ATL | WET | 1,93 (2,11) | 0,75 (0,65) | 0,11 (0,15) | urban (urban) |
| | DRY | 1,77 (1,94) | 1,00 (1,05) | -0,001 (0,006) | dust (dust/urban) |
| | BB | 1,80 (1,81) | 1,23 (1,08) | 0,008 (0,01) | dust/BB (urban) |
| APO | WET | 2,15 (2,04) | 0,84 (0,82) | 0,11 (0,10) | urban (urban) |
| | DRY | 1,39 (1,38) | 1,06 (1,10) | 0,006 (-0,02) | dust (dust) |
| | BB | 1,56 (1,61) | 1,14 (1,20) | -0,008 (-0,01) | dust/BB (dust/BB) |
| NES | WET | 2,05 (1,67) | 0,72 (0,66) | 0,13 (0,12) | urban (urban) |
| | DRY | 1,74 (1,83) | 1,06 (1,09) | -0,008 (0,003) | dust/urban (dust) |
| | BB | 1,89 (1,80) | 0,95 (1,07) | 0,002 (0,02) | dust/urban (urban) |

Table 2: Median aerosol Angström exponents of turbulent condition (stable condition) for each cluster and seasons measured at the CHC station and resulting aerosol types.

| Aerosol type | SAE | AAE | SSAAE |
|---|---|---|---|
| Dust | - | > 0,9 | [-0,05 ; 0,05] |
| Urban pollution | > 1,4 | < 0,9 | > 0,05 |
| Biomass burning | - | > 1,1 | [-0,05 ; 0,05] |

Table 3: Updated Angström exponent values expected for aerosol types at the CHC station.

l.127: "As a summary, Table 1 shows expected Angström exponent for dust, urban pollution and Biomass Burning particules according the different referenced works (Dubovik et al., 2002 ; Collaud Coen et al., 2004 ; Clarke et al., 2007 ; Russel et al., 2010). This information has to be taken with caution since source influences are expected homogeneous and have been reported from several regions."

l.482: "Table 2 summarizes the median Angström exponents measured at the CHC station for turbulent conditions (stable conditions in parenthesis). According to these values and as discussed above, aerosol types for the turbulent conditions (and stable conditions in parenthesis) are given."

l.525: "A new Angsrtöm exponent classification can then be defined for measurement at the CHC station and is reported Table 3. Thresholds are close to the ones proposed by previous works (Dubovik et al., 2002 ; Collaud Coen et al., 2004 ; Clarke et al., 2007 ; Russel et al., 2010) but adapted to CHC's instruments and particular atmospheric conditions."

The source region analysis performed in this work is puzzling. I am not sure how source regions have been distinguished. As an example C6 and C4 seem to overlap on Fig 4b. The same holds for source regions C1 and C2. There is a second graph in the lower left corner of Fig 4b for which I could not find any explanation. What is this graph about and how it is different than the main one of Fig4b? Please improve the caption of Fig 4 to include all information so that the reader can decipher the plots easily. Some of the info required to do so are found in the text but definitely the info provided is not enough. Since you have performed this analysis for 4 years, the individual trajectories for each source region should be shown in a separate (for each source region) graph in the Appendix. Please also add another plot showing the average trajectories for each source region on a map. Hysplit has this ability to produce average trajectories and so other software that are free to use. Personally I would recommend that the average trajectories graph for each cluster to be included in Fig 4. However it is not mandatory.

**Answer:** Cluster definition is based on the statistical method described by Borge et al. (2007) now better described in the text. Hence, the six clusters correspond to the main directions of the BTs as shown by the map figure 4. C1 and C2 show clear difference between their main origins (less transparent cells in Figure 4a). About C4 and C6, the difference is much on the distance range from CHC station which can be seen on the zoom in on the lower left corner. C4 corresponds to shorter distance (within 100 km) and C6 corresponds to longer distance. Maps of the first 10% BTs of each cluster and selected for the study are added in the appendix and clearly helps to visualize these features.

**Modifications:**

Figure 4 has been improved according RC1 recommendations. Scales have been added and geographical references improved.

[Figure]

[Figure]

Figure 4: a. Trajectory frequency plot in La Paz region in Bolivia, with a scale of 500 km and 50 km in the lower left corner centered on CHC station. b. Monthly variation of the percentage of back-trajectories (BTs) for each cluster.

Appendix A1:

[Figure]

**Figure A1: Selected 96-hours back-trajectories for the six clusters obtained from the Borge et al. (2007) method. The black line corresponds to the main back-trajectory.**

l.554: "For each hour of the period of the study, nine back-trajectories have been used to describe the mean influence at Chacaltaya station. The nine BTs start within a square of 2 km by 2 km around the station. The mean BT has been calculated from these nine BTs and generated every hour from January 2012 to December 2015. Clusters are defined according the Borge et al. (2017) method using a two-stage technique (based on the non-hierarchical K-means algorithm). The Borge et al. (2007) method allows to attribute to each mean BT a fraction of each cluster according to their time residence into the cluster and their distance from the CHC station. Hence, BTs are sorted according to their representativeness in each cluster. The first 10% of them are used in the present study and are reported in Fig. (A1)."

l.301: "Finally, cluster 6 (NES) has properties close to cluster 1 but with less influence from the Amazonian Basin and close to cluster 4 but with aerosol sources further from CHC station (> 100 km)."

There is a problem with the term ε in Eq 11. Is sa and ca are the sine and cosine of the same angle then by definition ε=0 regardless. This is due to the well known formula of cos2θ+sin2θ=1. I suspect a typo.

**Answer:** $s_a$ and $c_a$ are the average value of $\sin\theta$ and $\cos\theta$ in the 15 minutes time interval. As described in Yamartino et al. (1984), $s_a \neq \sin\theta_a$, $c_a \neq \cos\theta_a$ and $s_a{}^2 + c_a{}^2 \leq 1$.

**Modifications:**

l.231: Definition of ε has been corrected : " $\varepsilon = \sqrt{1 - (s_a^2 + c_a^2)}$ "

l.232: definition of sa and ca have also been corrected : "with the averages $s_a = \frac{1}{N}\sum_{i=1}^{N}\sin\theta_i$ and $s_a = \frac{1}{N}\sum_{i=1}^{N}\cos\theta_i$ of N the number of horitontal wind direction (θi) recorded in 15 minutes."

> In addition if σθ corresponds to the 15 minute average wind direction as stated in Line 179 what is the θ(15), θ(30), θ(45), θ(60) of Eq. 10. I thought they denoted different time intervals. Please spend some effort to explain further how the classification shown in Fig.2 is performed. In other words explain further what is discussed in Lines 181 and 182.

**Answer:** $\sigma^2_{\theta(15)}$ corresponds to the squared standard deviation of the horizontal wind direction calculated every 15 minutes. These quantities are later hourly averaged.

Indeed, atmospheric layer definitions are largely discussed on aerosol and dynamic studies, and according to the method used, the definitions of the layers are slightly different. In the present study, because we use atmospheric stability as tracer for atmospheric layers, it is more appropriate to use the two different regime "stable conditions" and "turbulent conditions".

**Modifications:**

In the full document, "layer" has been replaced by "condition" when needed, and SL – TL has been replaced by SC – TC (Stable and Turbulent Conditions).

l.220: "In addition, a residual layer can also be present at CHC station during nightime, resulting from low dispersion of the daytime convection. Because no clear distinctions between the mixing, the free tropospheric, and the residual layers can be strictly obtained from in-situ measurements only, the present dataset recorded at Chacaltaya station is separated in terms of stability conditions (turbulent and stable)."

l.227: "This method is based on the hourly averaged value of the standard deviation of the horizontal wind direction ($\sigma_\theta$ in Eq. 10) calculated every 15 minutes"

l.229: "with $\sigma_{\theta(15)}$ the standard deviation of the horizontal wind direction calculated on the first 15 minutes of every hour, and $\sigma_{\theta(60)}$ the last 15 minutes of every hour."

l.237: "As described in Rose et al. (2017), the classification depends also on the $\sigma_\theta$ value in the 4-hour time interval across the time of interest. Interface cases correspond to unclassified data which mainly show a high variability of the standard deviation between the two categories of dynamic. For clarity, the interface cases are excluded from the dataset in the rest of the paper."

> Despite that most of this work relates to phenomenology, there are two important findings. These are the very low AAE reported during the wet season and the linear relationship between SSAAE and AAE observed during the wet and dry seasons. I am wondering if such low AAE have been reported elsewhere in literature. Please discuss. Can the authors provide an explanation on the linear relationship observed in Fig9a?

**Answer:** The present study shows AAE reaching 0.5. These values are also observed by Russel et al. (2010) and corresponds to "urban industrial" impacts. AAE values from 1,5 to 3 correspond to dust particles.

The linear relationship between SSAAE and AAE can mainly be explained by a similarity on the sensibility to the two properties to two aerosol type. When air masses are mainly influenced by urban particles, especially during the wet season for every clusters, AAE values are close to 1 (or lower than BB influences) and SSAAE values are higher than 0. In the other hand, when air masses are mainly influenced by dust and BB particles (dry season for every clusters), AAE values are close to 2 (or much higher than during urban influences) and SSAAE values are close or below 0. A mixture between these two aerosol types will lead to intermediate values located in line with them.

**Modifications:**

l.445: "As shown in Fig. 5, low AAE values, especially during the wet season, can be explained by important reduction of dust and less biomass burning particles due to more efficient removal."

l.450: "Thus, the wet season presents positive SSAAE and AAE close or lower than 0.9, while dry season and BB period present SSAAE close to 0 and AAE higher than 0.9. A linear relationship between AAE and SSAAE values is observed and illustrates that mainly urban emissions drive aerosol particle properties during the wet period, and that mainly dust emissions drive aerosol particle properties during the dry season and the BB period."

There is a typo in the caption of Appendix Fig A1. Aerosol should probably be absorption and for the entire dataset instead of the all dataset

**Answer:** True

**Modifications:**

"Figure A2: Weekly variation of the Absorption Angström Exponent (AAE) for the whole dataset from 2012 to 2015. The medians and their 25th and 75th percentiles from Sundays to Saturdays are represented. "

**References:**

Borge, R., Lumbreras, J., Vardoulakis, S., Kassomenos, P. and Rodríguez, E.: Analysis of long-range transport influences on urban PM 10 using two-stage atmospheric trajectory clusters, Atmos. Environ., 41(21), 4434–4450, 2007.

Clarke, A., McNaughton, C., Kapustin, V., Shinozuka, Y., Howell, S., Dibb, J., Zhou, J., Anderson, B., Brekhovskikh, V., Turner, H. and Pinkerton, M.: Biomass burning and pollution aerosol over North America: Organic components and their influence on spectral optical properties and humidification response, J. Geophys. Res. Atmospheres, 112(D12), D12S18, doi:10.1029/2006JD007777, 2007.

Collaud Coen, M., Weingartner, E., Apituley, A., Ceburnis, D., Fierz-Schmidhauser, R., Flentje, H., Henzing, J. S., Jennings, S. G., Moerman, M., Petzold, A., Schmid, O. and Baltensperger, U.: Minimizing light absorption measurement artifacts of the Aethalometer: evaluation of five correction algorithms, Atmospheric Measurement Techniques, 3(2), 457–474, doi:10.5194/amt-3-457-2010, 2010.

Dubovik, O., Holben, B., Eck, T. F., Smirnov, A., Kaufman, Y. J., King, M. D., Tanre, D. and Slutsker, I.: Variability of absorption and optical properties of key aerosol types observed in worldwide locations, J. Atmospheric Sci., 59, 590–608, doi:Review, 2002.

Rose, C., Sellegri, K., Moreno, I., Velarde, F., Ramonet, M., Weinhold, K., Krejci, R., Andrade, M., Wiedensohler, A. and Ginot, P.: CCN production by new particle formation in the free troposphere, Atmospheric Chemistry and Physics, 17(2), 1529–1541, 2017.

Russell, P. B., Bergstrom, R. W., Shinozuka, Y., Clarke, A. D., DeCarlo, P. F., Jimenez, J. L., Livingston, J. M., Redemann, J., Dubovik, O. and Strawa, A.: Absorption Angstrom Exponent in AERONET and related data as an indicator of aerosol composition, Atmospheric Chem. Phys., 10, 1155–1169, doi:10.5194/acp-10-1155-2010, 2010.

Yamartino, R. J.: A comparison of several "single-pass" estimators of the standard deviation of wind direction, J. Clim. Appl. Meteorol., 23(9), 1362–1366, 1984.

---

## Author Comment (AC2) · 22 Oct 2019

First, authors would like to thank the reviewer for his/her constructive and detailed suggestions. Their additions to the paper helping us to improve significantly the clarity and the precision of the method used and the robustness of the results.

The atmospheric stability is used as a tracer to differentiate the atmospheric layers. As recommended, Stable Layer (SC) and Turbulent Layer (TL) are have been replaced by Stable Conditions (SC) and Turbulent Conditions (TC). This was also requested by other 2 reviewers.

In the following, we provide answers to the reviewer's comments and list modifications made in the manuscript.

This paper provides an interesting overview of the aerosol particle properties observed at the high mountain station Chacaltaya in the South America (Cordillera Real). The topic is of interest for ACP and the paper is generally well written. The scientific approach is sufficiently robust and the presentation of data and results is fair. Nevertheless, some points should be better addressed before publications. In particular, the authors should better discuss the caveats related with the back-trajectories analysis as well as provide more details and information about the experimental methodologies (e.g., no information about data generation, uncertainty characterization are provided).

In the following you can find my specific comments. ——————————————————— ——————— ————————————————

Abstract. In the last sentence, the authors claimed that "CHC provides first evidences of impact of emission from Amazonian basin far away from their source". Be more specific. Which "far away" means? Please, in the site description provide distance of CHC from Amazonian basin.

**Answer:** The CHC station is described as a background GAW site and is located 17 km from the first urban area, and around 300 km from fire area in Bolivia (Carmona-Moreno et al. 2005, Giglio et. al 2013). The Rondonia region where a significate deforestation process is reported every year is located 800 km from CHC station. In addition, CHC is the highest GAW station in the world and unique free tropospheric conditions can be obtained in the region where particles and gases have residence time of several months and be transported over large distances.

**Modifications:**

l.56: "From this analysis, long-term observations at CHC provides the first direct evidence of the impact of Biomass Burning emissions of the Amazonian basin and urban emissions from La Paz area on atmospheric optical properties to a remote site all the way to the free troposphere."

l.148: "[...] (Carmona-Moreno et al. 2005, Giglio et. al 2013). Indeed, the closest region where large areas are affected by biomass burning activities is the Bolivian Amazonia (Beni, Santa Cruz, north of La Paz departments) located ca. 300 km from the station, north and eastward from the Andes mountain range."

Line 71: please provide wavelengths.

**Answer:** Husar et al., (2000) reports aerosol extinction coefficients values related to visibility. Their values are representative of the integrated visible range. In this condition, it may not be reliable to

compare those aerosol extinction coefficients with the Chacaltaya measurements. However, the study gives an interesting analysis of the impact of Amazonian fires at different altitudes and different distances to the main fire activities.

**Modifications:**

l.84: "Between the wet season and the biomass burning season, Schafer et al. (2008) show an increase of Aerosol Optical Depth by a factor of 10 from AERONET sites in southern forest region and the Cerrado region and, by a factor of 4 in the northern forest region."

l.88: "The study reports a spatial pattern of the visibility between 100 and 200 Mm-1 over the Amazon Basin. However, values can reach 600 Mm-1 at Sucre station (2903 m above sea level, hereafter abbreviated as "a.s.l."), 1000 Mm-1 at Vallegrande (1998m a.s.l.) and 2000 Mm-1 at Camiri (792 m a.s.l.) during BB period. Even the study clearly shows impacts of Amazonian activities at different altitudes and long distances, only few studies report long time period of aerosol optical properties."

l.96: "These extremely high coefficients are due to the proximity to BB sources for FNSA station and its very low altitude."

Line 92 – 105: this section ìs hard to follow. I would recommend to add a table with the different threshold values for each type of particles (dust, pollution, biomass burning) for the different Angstrom exponents (AAE, SAE, SSAE).

**Answer:** Three tables have been added to the text which summarize information from the text. The Table 1 summarizes ranges of Angström exponent for the different aerosol types. Table 2 details the median values of the Angström exponent for each cluster, season and atmospheric stability. Table 3, on the conclusion, suggests a new Angtsröm exponent definition for the different aerosol types.

**Modifications:**

| Aerosol type | SAE | AAE | SSAAE |
|---|---|---|---|
| Dust | Close to 1 | Close to 1 | Below 0 |
| Urban pollution | Close to 2 | Close to 1 | Higher than 0 |
| Biomass burning | | Close to 2,1 | |

Table 1: Expected aerosol type and their optical properties for each cluster according season and atmospheric stability.

| Cluster | season | SAE | AAE | SSAAE | Aerosol types |
|---------|--------|-----|-----|-------|---------------|
| NA | WET | 2,04 (1,42) | 0,58 (0,56) | 0,18 (0,15) | urban (dust/urban) |
| | DRY | 1,91 (1,80) | 1,00 (1,01) | 0,01 (0,004) | urban (dust) |
| | BB | 1,92 (1,87) | 1,10 (1,26) | 0,03 (0,02) | dust/BB (dust/BB) |
| SA | WET | 1,2 (1,40) | 0,74 (0,68) | 0,11 (0,11) | urban (urban) |
| | DRY | 1,69 (1,70) | 1,04 (0,96) | 0,02 (0,03) | dust (dust) |
| | BB | 2,16 (2,02) | 1,23 (1,20) | 0,005 (0,01) | BB (BB) |
| LP | WET | 1,71 (2,09) | 0,86 (0,82) | 0,08 (0,10) | urban (urban) |
| | DRY | 1,64 (1,74) | 1,05 (1,07) | 0,02 (-0,01) | urban (dust/urban) |
| | BB | 1,49 (1,93) | 1,09 (1,29) | -0,02 (-0,02) | dust (dust/BB) |
| ATL | WET | 1,93 (2,11) | 0,75 (0,65) | 0,11 (0,15) | urban (urban) |
| | DRY | 1,77 (1,94) | 1,00 (1,05) | -0,001 (0,006) | dust (dust/urban) |
| | BB | 1,80 (1,81) | 1,23 (1,08) | 0,008 (0,01) | dust/BB (urban) |
| APO | WET | 2,15 (2,04) | 0,84 (0,82) | 0,11 (0,10) | urban (urban) |
| | DRY | 1,39 (1,38) | 1,06 (1,10) | 0,006 (-0,02) | dust (dust) |
| | BB | 1,56 (1,61) | 1,14 (1,20) | -0,008 (-0,01) | dust/BB (dust/BB) |
| NES | WET | 2,05 (1,67) | 0,72 (0,66) | 0,13 (0,12) | urban (urban) |
| | DRY | 1,74 (1,83) | 1,06 (1,09) | -0,008 (0,003) | dust/urban (dust) |
| | BB | 1,89 (1,80) | 0,95 (1,07) | 0,002 (0,02) | dust/urban (urban) |

Table 2: Median aerosol Angström exponents of turbulent condition (stable condition) for each cluster and seasons measured at the CHC station and resulting aerosol types.

| Aerosol type | SAE | AAE | SSAAE |
|--------------|-----|-----|-------|
| Dust | - | > 0,9 | [-0,05 ; 0,05] |
| Urban pollution | > 1,4 | < 0,9 | > 0,05 |
| Biomass burning | - | > 1,1 | [-0,05 ; 0,05] |

Table 3: Updated Angström exponent values expected for aerosol types at the CHC station.

l.127: "As a summary, Table 1 shows expected Angström exponent for dust, urban pollution and Biomass Burning particles according the different referenced works (Dubovik et al., 2002 ; Collaud Coen et al., 2004 ; Clarke et al., 2007 ; Russel et al., 2010). This information has to be taken with caution since source influences are expected homogeneous and have been reported from several regions."

l.482: "Table 2 summarizes the median Angström exponents measured at the CHC station for turbulent conditions (stable conditions in parenthesis). According to these values and as discussed above, aerosol types for the turbulent conditions (and stable conditions in parenthesis) are given."

l.525: "A new Angtsröm exponent classification can then be defined for measurement at the CHC station and is reported Table 3. Thresholds are close to the ones proposed by previous works (Dubovik et al., 2002 ; Collaud Coen et al., 2004 ; Clarke et al., 2007 ; Russel et al., 2010) but adapted to CHC's instruments and particular atmospheric conditions."

Line 103: please correct "bellow"

**Modifications:** "bellow" has been corrected by "below"

Line 135: please clearly state which kind of compensation must be applied to aethalometer data.

**Answer:** Weingartner et al. (2003) correction is applied in order to compensate the multi-scattering effects and the loading effects on the aethalometer's filters. The correction is performed by adjusting the f factor showed in equation 3.

**Modifications:**

l.170:"Aethalometer measurements were compensated for multi-scattering effects and loading effects (or shadowing effects) following the method described by Weingartner et al. (2003) briefly explained below."

Line 140. was the mass coefficient provided by the manufacturer independently assessed and validated by others? If yes, provide references, if not, provide adequate comments

**Answer:** The mass coefficient given by the manufacturer allows us to convert BC concentrations to attenuation coefficients as derived from the Mie theory for small uniform spheres. Hence, the mass coefficient is inversely proportional to the optical wavelength as follow: $\sigma= 14625 / \lambda$, which corresponds to sigma values between 15 and 40 $m^2\ g^{-1}$ for wavelengths between 370 and 950 nm. Liousse et al. (1993) demonstrated highly variable mass coefficients depending on aerosol type and age. Saturno et al. (2017) recently demonstrate agreement between aethalometer corrections using manufacturer's sigma coefficients and other instruments as MAAP and a multiple-wavelength absorbance analyser (MWAA). Collaud Coen et al. (2010) also demonstrate Weingartner's correction as a good compromise comparing to Schmid's et al. (2006) correction and Arnott's et al. (2005) correction.

**Modifications:**

l.176: "with $\sigma m$ the mass coefficients given by the instrument's instructions (The Aethalometer, A.D.A. Hansen, Magee Scientific Company, Berkley, California, USA) and based on the Mie theory. $\sigma m$ strongly depends on the aerosol type and age (from 5 to 20 $m^2$ g-1, Liousse et al., 1993). However, the manufacturer values (14625 nm m2 g-1 $\lambda$-1) have been recently validated in a comparison study between ifferent aethalometer corrections (Collaud Coen et al. 2010 ; Saturno et al., 2017)."

The method for deriving the absorption coefficient it is not clear. Equation 2, what is C.R(ïA˛ˇn)?. Equation 3 it is also not clear: please describe the contribution of each member/factor. What ln(10%) and ln(50%) represent? Why the factor R should be adjusted? What do you mean for "spot" change? Why the absorption coefficient should be the same before and after the spot change?

**Answer:** "C.R(ïA˛ˇn)" is probably due to format issues and should be C.R($\lambda$). Then, C is defined as the calibration factor constant with wavelength, and R another calibration factor which does depend on the wavelength.

The aethalometer instrument permits to obtain aerosol absorption coefficients from optical measurement of aerosol trapped on a filter. Every 5 minutes, the spot on the filter band is changed in order to reduce loading effects. Hence, absorption coefficients from one spot to the other should not

change significantly, then the median ratio between two successive absorption retrievals should be less than 1.

The method to derive absorption coefficients and correct data from aethalometer issues (multi-scattering and loading effect) is briefly described in this paper. However, the method is described in details in Weingartner et al. (2003) and largely used in other several studies (Bond et al., 2006, 2013 ; Rose et al., 2015 ; Andreae and Gelencsér, 2006 ; Zotter et al., 2017 ; Rajesh and Ramachandran, 2018). If C allows to correct multi-scattering effects linked to filter properties, R has to be adjusted in order to correct the loading effect and is related to wavelength and aerosol properties trapped on the filter. In Weingartner et al. (2003), we can find the description of every member. f is the filter loading effect compensation parameter and represents the slope of the curve of R as function of $\ln(\sigma_{atn})$ for a $\sigma_{atn}$ change from 10% to 50%.

**Modifications:**

l.167: "Every 5 minutes, the spot on the filter band is changed in order to reduce loading effects."

l.170: "Aethalometer measurements were compensated for multi-scattering effects and loading effects (or shadowing effects) with the method described by Weingartner et al. (2003) and briefly explained below."

l.176: "with $\sigma_m$ the mass coefficients given by the instrument's instructions (The Aethalometer, A.D.A. Hansen, Magee Scientific Company, Berkley, California, USA) and based on the Mie theory. $\sigma_m$ strongly depends on the aerosol type and age (from 5 to 20 m² g-1, Liousse et al., 1993). However, the manufacturer values (14625 nm m2 g-1 λ-1) have been recently validated in a comparison study between aethalometer corrections (Collaud Coen et al., 2010 ; Saturno et al., 2017)."

l.182: "with C = 3.5 a calibration factor linked to multiple-scattering and assumed constant according wavelengths (GAW Report No. 227), and R, a calibration factor which depends on aerosol loading on the filter and aerosol optical properties, calculated as: [...]"

l.186: "where f is the filter loading effect compensation parameter and represents the slope of the curve of R as function of $\ln(\sigma_{atn})$ for a $\sigma_{atn}$ change from 10% to 50%"

Equation 4: what "mEBC" is? Does QEBC is equal to QBC reported by line 148?

**Answer:** EBC corresponds to Equivalent Black Carbon. Because Black Carbon concentrations are obtained from optical measurements, and assumptions on mass absorption cross-section coefficients have to be included, it seems more appropriate to use Equivalent Black Carbon. $Q_{BC}$ should be $Q_{EBC}$ everywhere in the manuscript.

**Modifications:**

l.191: "A mass absorption cross-section $Q_{EBC}$ = 6.6 $m^2.g^{-1}$ at 670 nm is used to determine Equivalent Black Carbon mass concentrations"

Line 155: the authors stated that the aethalometer measurement at 635 mm is unstable. Quantitatively, what does this mean? Do you are able to provide threshold value that other users can apply to evaluate if their own measurements are unstable? In general, which QA/QC framework/procedures did you apply to all the suite of measurement discussed in this work? Please

describe air inlet system and calibration strategies for the considered instrumentations (i.e. nephelometer, aethalometr, MAAP). Please quantify uncertainties related to each of these measurements.

**Answer:**

Indeed, nephelometer measurements at 635 nm do show unstable values during several days. The figure below shows an example of these unstable periods, here for the scattering coefficients at 450, 525 and 635 nm between the 7th of September to the 9th of September 2012. The wavelength dependence of the scattering coefficients suddenly changes the 7th of September around 6 pm and the 635 nm channel keeps high values for several months. Level 2 data used in this work are in the EBAS database and consequently these data are controlled and opportunely flagged when issues were observed. The authors decided to only use unchanged 450 and 525 nm channels to analyse aerosol optical properties at Chacalataya.

[Figure]

Figure 1: Temporal serie of the scattering coefficients at 450, 525 and 635 nm retrieved from the nephelometer at Chacaltaya station between the 7th of September at 00 am to 9th of September 2012 at 00 am.

From optical property measurements, the full dataset between the 3rd of January 2012 to 30th of November 2015 has been used. Every in-situ instruments of the station are located downstream a Whole Air Inlet and a dryer.

The nephelometer instrument was periodically calibrated using $CO_2$ as span gas and frequent zero adjusts were performed, following the procedure described in Ecotech manual (2009). The uncertainty of the Aurora 3000 is given in the user manual to be 2,5 %.

The air-flow and the absorption cross-section of the aethalometer instrument are calibrated. The method is detailed on the Magee Scientific manual (REF). Accuracy of attenuation coefficients are around 5%.

The MAAP instrument is automatically calibrated for air-flow, temperature and pressure according to the instruction manual (Model 5012 Instruction Manual) and described in Petzold and Schönlinner, 2004). According to them, uncertainty of the absorbance is 12%.

**Modifications:**

l.156: "In-situ instruments of the station operated behind a Whole Air Inlet equipped with an automatic dryer (activated above 90% RH) ."

l.158: "In the current study, the full dataset of in-situ optical measurements has been used between January 2012 and December 2015."

l.169: "Sensor calibration is performed automatically and an uncertainty of 5 % on attenuation coefficients is given by the constructor."

l.190: "According to Petzold and Schönlinner (2004), uncertainty of the absorbance is 12%."

l.198: "The nephelometer instrument is calibrated using $CO_2$ as span gas and frequent zero adjusts were performed, following the procedure described in Ecotech manual (2009). The uncertainty of the Aurora 3000 is given in the user manual to be 2,5 %."

Also equation 10 is not clear: what $\sigma\ddot{\imath}'2$ (x) with X=12,30,45,60 represent? I think that the vocabulary used by the authors can be misleading. More than layers this methodology can be able to discriminate turbulent versus stable (or more stable) conditions at the measurement site. Please change nomenclature.

**Answer:** $\sigma_{\theta(15)}^2$ corresponds to the squared standard deviation of the horizontal wind direction calculated every 15 minutes. Hence, the standard deviation of the horizontal wind direction is calculated every 15 minutes and then hourly averaged.

Indeed, atmospheric layer definitions are largely discussed on aerosol and dynamic studies, and according to the method used, the definitions of the layers are slightly different. In addition, a residual layer can also be present at CHC station during nightime, resulting from low dispersion of the daytime convection. In the present study, because we use atmospheric stability as tracer for atmospheric layers, it is more appropriate to use the two different regime "stable conditions" and "turbulent conditions".

**Modifications:**

In the full document, "layer" has been replaced by "condition" when needed, and SL – TL has been replaced by SC – TC (Stable and Turbulent Conditions).

l.220: "In addition, a residual layer can also be present at CHC station during nightime, resulting from low dispersion of the daytime convection. Because no clear distinctions between the mixing, the free tropospheric, and the residual layers can be strictly obtained from in-situ measurements only, the present dataset recorded at Chacaltaya station is separated in terms of stability conditions (turbulent and stable)."

l.227: "This method is based on the hourly averaged value of the standard deviation of the horizontal wind direction ($\sigma_\theta$ in Eq. 10) calculated every 15 minutes"

l.229: "with $\sigma_{\theta(15)}$ the standard deviation of the horizontal wind direction calculated on the first 15 minutes of every hour, and $\sigma_{\theta(60)}$ the last 15 minutes of every hour."

Line 189: why was the residual layer excluded by the analysis? Does this mean that the residual layer conditions are embedded in what the author defined as "stable" layers? Please, better specify this point since this can have implications for the interpretation of results.

**Answer:** The residual layer is finally reported as stable condition. Hence, the interface cases do not take into account the residual layer but cases which cannot be attributed to stable or turbulent conditions. Indeed, these cases correspond to unclear dynamical conditions with high variability of the

standard deviation of the horizontal wind direction. In order to analyse aerosol properties in stable and turbulent conditions only, the interface measurements have to be excluded.

**Modifications:**

**l**.220: "In addition, a residual layer can also be present at CHC station during nightime, resulting from low dispersion of the daytime convection. Because no clear distinctions between the mixing, the free tropospheric, and the residual layers can be strictly obtained from in-situ measurements only, the present dataset recorded at Chacaltaya station is separated in terms of stability conditions (turbulent and stable)."

l.236: "Interface cases correspond to unclassified data which mainly show high variability of the standard deviation between the two categories of dynamic. As described in Rose et al. (2017), the classification depends also on the $\sigma_\theta$ value in the 4-hour time interval across the time of interest. Interface cases correspond to unclassified data which mainly show a high variability of the standard deviation between the two categories of dynamic. For clarity, the interface cases are excluded from the dataset in the rest of the paper."

l.249: "Black spots represent undefined cases (or interface) due to a fluctuating classification within the 1-hour time window."

l.500: "Even TC is usually attributed to mixing layer, SC can be undoubtedly attributed to free tropospheric or residual layers."

Line 193: I think that "morning" must be changed by "night"

**Answer:** True.

**Modifications:**

l.250: "This 3-day example shows that SLSC conditions are mostly observed during night when the convective effect of the previous day is already dissipated and no convective effect of the current day is present."

Line 204: what BT set is used for the cluster analysis (12 hours or 96 hours)? Why different TRJ lengths were considered/calculated? Is the trajectory calculation set-up changing for the 12 and the 96 hours BTs? The authors did not provide any indication about the meteorological files (which are? Which horizontal and vertical resolution?) used for BT calculation nor about calculation set-up (which starting heights? single or multiple starting points around the station locations). The resolution of the input metro files is particularly important in this mountain region, I guess. Please comment on that and provide caveats about the effective reliability of trajectories in this region. This point is critical for interpretation of results.

**Answer:**

Indeed, only 96 hours back-trajectories are used in cluster analysis. 96-hours BTs are more appropriate in this region to analyse long range transports in particular from the Amazonian forest.

Hysplit trajectories has been generated using WRFd04 data and the kinematic method with ERA-interim data as boundary conditions. This dataset presents the best topographic resolution for this region with spatial resolution of 1.06x1.06 km and give an altitude of Chacaltaya station of 5058 m

a.s.l. (true altitude 5240 m a.s.l.). The WRF dataset presents 28 atmospheric levels of pressure given every 6 hours. For this study, 96-hours BTs are generated every hour starting at 9 locations around the Chacaltaya station (Table 4 below).

| Latitude (degrees) | Longitude (degrees) | Altitude (m a.s.l.) |
|---|---|---|
| -16,36 | -68,14 | 4852 |
| -16,35 | -68,14 | 4918 |
| -16,34 | -68,14 | 4883 |
| -16,36 | -68,13 | 5000 |
| **-16,35** | **-68,13** | **5058** |
| -16,34 | -68,13 | 4965 |
| -16,36 | -68,12 | 5042 |
| -16,35 | -68,12 | 5043 |
| -16,34 | -68,12 | 4936 |

Table 4: Positions of the 9 stating location of BTs calculations. Chacaltaya station is bolded.

**Modifications:**

l.264: "WRFd04 dataset has been used to generate BTs every hour, starting at nine locations at less than 1 km around the Chacaltaya station (within a square of 2x2 km around the station). This dataset presents the best topographic resolution for this region with spatial resolution of 1.06x1.06 km, and 28 pressure levels."

Figure 4a is hard to understand and the comparison among the different cluster is challenging. Maybe, it can help to use a stack bar plot with 1 bar for each single month composed by the contribution from each different single cluster.

Also Figure 4b is difficult. The geographical boundaries are not clear at all. The same is true for the topographic features. Most of the locations listed in the legend are meaningfulness for readers not used with the region (are these villages, cities, regions?). For these reasons, it should be strongly improved.

**Answer:** Figures 4a and 4b have been improved. Indeed, stack bars allow to compare contribution of each cluster every month and is better appropriate to details given in the text. As recommended by RC1, trajectories of each cluster have also been added to Appendix A to illustrate results of the cluster analysis method.

**Modifications:**

Figure 4a and 1b have been improved.

legend Figure 4: "a. Trajectory frequency plot in La Paz region in Bolivia, centered on Chacaltaya station. ba. Monthly variation of the percentage of back-trajectories (BTs) for each cluster."

Appendix A added with the 10% first BTs more representative of each cluster.

Line 212: sentence starting with "Thus, for each cluster,..." isnt.t clear: what do you mean for "events"? " When the cluster have the most influence": what does it mean?

**Answer:** In order to obtain aerosol optical properties of each cluster, only a part of the back-trajectories have been selected. One BT is selected if its contribution to one cluster is high enough.

Hence, this can be obtained by selecting the first 10% of the BTs which are the most representative of each cluster. The describing paragraph has been improved to explain this selection of BTs.

**Modifications:**

l.268: "The Cluster Analysis method used in this study is described in Borge et al. (2007) and based on the Euclidean geographical coordinates distance and given time intervals. Figure (4a) shows the trajectory frequency plot. The opacity of each pixel is proportional to the number of BTs passing through each grid cell. Clusters are defined by using a two-stage technique (based on the non-hierarchical K-means algorithm). Six clusters have been found around the Chacaltaya station. Hence, a fraction of each cluster is assigned to each BT, and is calculated according the residence time in each cluster and their distance from the reference location (the Chacaltaya station). In order to obtain aerosol optical properties of each cluster, only a part of the back-trajectories have been selected. One BT is selected if its contribution to one cluster is high enough. For each cluster, the first 10% of the BTs have been selected by demonstrating the highest contribution to one any cluster. This firsts 10% of BTs related to each clusters and their mean paths are shown in Appendix A1

Line 263: Since the extinction is the sum of absorption and scattering, and scattering » absorption, the similarity between extinction and scattering is trivial.

**Answer:** This sentence permits to show that the extinction coefficient is mainly driven by the scattering property. This is regularly the case and can be deleted here.

**Modifications:**

l.333: "[...] and follows a seasonal variation that is very similar to the one of the scattering coefficient" has been deleted.

Line 377 - 376: please better explain. in which way the AAE values are impacted by the aethalometer variability. Do you mean that the uncertainty of aethalometer is enhanced during wet season? For which reason? How this impact results robustness (please discuss in the conclusions)?

**Answer:** The absorption Angtström exponent is calculated according equation 7 by using wavelength dependency of the absorption coefficients measured by the Aethalometer. As described in the description of the Aethalometer correction, absorption measurements can be biased by an important aerosol loading on filters and multi-scattering effects. However, two physical ways can explain these low AAE values: a reduction of dust and less biomass burning particles due to more efficient removal (higher hygroscopicity of BB particles). The Figure shows the frequency plot of the AAE for the entire period. Results are centered around 1 but vary from -0.5 to 2.

[Figure]

Figure 2 : Frequency plot of the Absorption Angström Exponent for the full period of the study.

**Modifications:**

l.445: "As shown in Fig. 5, low AAE values, especially during the wet season, can be explained by important reduction of dust and less biomass burning particles due to more efficient removal."

Line 401: the decrease of urban particle influence within air-masses from LaPaz during "turbulent" conditions (in which I expect more efficient transport from the lower layers to CHC) is rather surprising. Does this indicate some inaccuracies in the local TRJ calculation or in the turbulent conditions identification?

**Answer:** Capturing the local atmospheric stability is still highly challenging in this region due to the complex topography and high altitude. In addition to these inaccuracies, and as mentioned in the text, dust particles largely generated in this region can also affect measurements in turbulent conditions measured at Chacaltaya station and thus, decrease SAE values. It can also be noticed that SSAAE are still representative of urban influences with values above 0. This indicates the main influence of urban particles from LP cluster in both stability conditions but with a moderate addition of dust influence in the turbulent condition cases.

**Modifications:**

l.474: "In addition to urban influences, during the BB period and the wet season, LP air masses are also affected by dust particles, especially in the TC, with significantly lower SAE values in the TL."

Line 421:I agree that the transport to higher troposphere layers was supported but the spread over long-range (please, quantitatively specify what do you mean for longrange) is more a (reliable) hypothesis.

**Answer:** The long-range transport from the aerosol sources still a challenge to define since no detailed analysis is available about aerosol source locations. However, for BB emissions, the first intensive biomass burning source is located at 300 km fare from Chacaltaya station.

**Modifications:**

l.148: "[...] (Carmona-Moreno et al. 2005, Giglio et. al 2013). Indeed, the closest region where large areas are affected by biomass burning activities is the Bolivian Amazonia (Beni, Santa Cruz, north of La Paz departments) located ca. 300 km from the station, north and eastward from the Andes mountain range."

l.505: "The present study has hence demonstrated that BB particles are efficiently transported to the higher part of the troposphere (Stable conditions) and over long distances (more than 300 km long)."

Line 435: the authors concluded that an effect of dust is visible during the entre dry season. However, looking at figure 9, SSAAE is mostly >0 during the dry season which contradict this (see also line 103). Please comment and/or rephrase.

**Answer:** In this arid region, dust sources have a significant impact on ground based measurements especially when the station is in turbulent conditions which tend SSAAE values to be below 0. However, air masses are not purely influenced by dust particles and are always slightly impacted by urban or BB emissions. In these conditions, SSAAE values are close to 0. In addition, Figure 7 shows the large variability of SSAAE values for every season and atmospheric condition due to the complexity of the different contribution of each type of aerosol measured at CHC. Ealo et al. (2016) have also shown that SSAAE performance in detecting dust is related to the amount of fine particles.

**Modifications:**

l.521: "In addition to urban and BB influences, the wavelength dependence of the single scattering albedo (SSAAE) measured at CHC highlights a main dust influence during the entire dry season with SSAAE values close to 0."

**References:**

Arnott, W. P., Hamasha, K., Moosmüller, H., Sheridan, P. J. and Ogren, J. A.: Towards Aerosol Light-Absorption Measurements with a 7-Wavelength Aethalometer: Evaluation with a Photoacoustic Instrument and 3-Wavelength Nephelometer, Aerosol Science and Technology, 39(1), 17–29, doi:10.1080/027868290901972, 2005.

Andreae, M. O. and Gelencsér, A.: Black carbon or brown carbon? The nature of light-absorbing carbonaceous aerosols, Atmospheric Chemistry and Physics, 6(10), 3131–3148, 2006.

Bond, T. C., Habib, G. and Bergstrom, R. W.: Limitations in the enhancement of visible light absorption due to mixing state, Journal of Geophysical Research-Atmospheres, 111, doi:Article, 2006.

Bond, T. C., Doherty, S. J., Fahey, D. W., Forster, P. M., Berntsen, T., DeAngelo, B. J., Flanner, M. G., Ghan, S., Kärcher, B., Koch, D., Kinne, S., Kondo, Y., Quinn, P. K., Sarofim, M. C., Schultz, M. G., Schulz, M., Venkataraman, C., Zhang, H., Zhang, S., Bellouin, N., Guttikunda, S. K., Hopke, P. K., Jacobson, M. Z., Kaiser, J. W., Klimont, Z., Lohmann, U., Schwarz, J. P., Shindell, D., Storelvmo, T., Warren, S. G. and Zender, C. S.: Bounding the role of black carbon in the climate system: A scientific assessment, Journal of Geophysical Research: Atmospheres, 118(11), 5380–5552, doi:10.1002/jgrd.50171, 2013.

Carmona-Moreno, C., Belward, A., Malingreau, J.-P., Hartley, A., Garcia-Alegre, M., Antonovskiy, M., Buchshtaber, V. and Pivovarov, V.: Characterizing interannual variations in global fire calendar using data from Earth observing satellites, Global Change Biology, 11(9), 1537–1555, 2005.

Clarke, A., McNaughton, C., Kapustin, V., Shinozuka, Y., Howell, S., Dibb, J., Zhou, J., Anderson, B., Brekhovskikh, V., Turner, H. and Pinkerton, M.: Biomass burning and pollution aerosol over North America: Organic components and their influence on spectral optical properties and humidification response, J. Geophys. Res. Atmospheres, 112(D12), D12S18, doi:10.1029/2006JD007777, 2007.

Collaud Coen, M., Weingartner, E., Apituley, A., Ceburnis, D., Fierz-Schmidhauser, R., Flentje, H., Henzing, J. S., Jennings, S. G., Moerman, M., Petzold, A., Schmid, O. and Baltensperger, U.: Minimizing light absorption measurement artifacts of the Aethalometer: evaluation of five correction algorithms, Atmospheric Measurement Techniques, 3(2), 457–474, doi:10.5194/amt-3-457-2010, 2010.

Dubovik, O., Holben, B., Eck, T. F., Smirnov, A., Kaufman, Y. J., King, M. D., Tanre, D. and Slutsker, I.: Variability of absorption and optical properties of key aerosol types observed in worldwide locations, J. Atmospheric Sci., 59, 590–608, doi:Review, 2002.

Ealo, M., Alastuey, A., Ripoll, A., Pérez, N., Minguillón, M. C., Querol, X., and Pandolfi, M.: Detection of Saharan dust and biomass burning events using near-real-time intensive aerosol optical properties in the north-western Mediterranean, Atmos. Chem. Phys., 16, 12567–12586, https://doi.org/10.5194/acp-16-12567-2016, 2016.

Ecotech, Aurora 3000 User manual 1.3, November 2009.

Giglio, L., Randerson, J. T. and van der Werf, G. R.: Analysis of daily, monthly, and annual burned area using the fourth-generation global fire emissions database (GFED4), Journal of Geophysical Research: Biogeosciences, 118(1), 317–328, 2013.

Husar, R. B., Husar, J. D. and Martin, L.: Distribution of continental surface aerosol extinction based on visual range data, Atmos. Environ., 34(29–30), 5067–5078, doi:10.1016/S1352-2310(00)00324-1, 2000.

Liousse, C., Cachier, H. and Jennings, S. G.: Optical and thermal measurements of black carbon aerosol content in different environments: Variation of the specific attenuation cross-section, sigma (σ), Atmospheric Environment. Part A. General Topics, 27(8), 1203–1211, 1993.

Rajesh, T. A. and Ramachandran, S.: Black carbon aerosol mass concentration, absorption and single scattering albedo from single and dual spot aethalometers: Radiative implications, Journal of Aerosol Science, 119, 77–90, 2018.

Rose, C., Sellegri, K., Velarde, F., Moreno, I., Ramonet, M., Weinhold, K., Krejci, R., Ginot, P., Andrade, M. and Wiedensohler, A.: Frequent nucleation events at the high altitude station of Chacaltaya (5240 m asl), Bolivia, Atmos. Environ., 102, 18–29, 2015.

Russell, P. B., Bergstrom, R. W., Shinozuka, Y., Clarke, A. D., DeCarlo, P. F., Jimenez, J. L., Livingston, J. M., Redemann, J., Dubovik, O. and Strawa, A.: Absorption Angstrom Exponent in AERONET and related data as an indicator of aerosol composition, Atmospheric Chem. Phys., 10, 1155–1169, doi:10.5194/acp-10-1155-2010, 2010.

Saturno, J., Pöhlker, C., Massabó, D., Brito, J., Carbone, S., Cheng, Y., Chi, X. and Ditas, F.: Comparison of different Aethalometer correction schemes and a reference multi-wavelength absorption technique for ambient aerosol data, Atmos. Meas. Tech., 15, 2017.

Schafer, J. S., Eck, T. F., Holben, B. N., Artaxo, P. and Duarte, A. F.: Characterization of the optical properties of atmospheric aerosols in Amazônia from long-term AERONET monitoring (1993–1995 and 1999–2006), Journal of Geophysical Research: Atmospheres, 113(D4), doi:10.1029/2007JD009319, 2008.

Schmid, O., Artaxo, P., Arnott, W. P., Chand, D., Gatti, L. V., Frank, G. P., Hoffer, A., Schnaiter, M. and Andreae, M. O.: Spectral light absorption by ambient aerosols influenced by biomass burning  in the

Amazon Basin. I: Comparison and field calibration of absorption measurement techniques, Atmos. Chem. Phys., 6(11), 3443–3462, doi:10.5194/acp-6-3443-2006, 2006.

Weingartner, E., Saathoff, H., Schnaiter, M., Streit, N., Bitnar, B. and Baltensperger, U.: Absorption of light by soot particles: determination of the absorption coefficient by means of aethalometers, J. Aerosol Sci., 34(10), 1445–1463, doi:10.1016/S0021-8502(03)00359-8, 2003.

Zotter, P., Herich, H., Gysel, M., El-Haddad, I., Zhang, Y., Močnik, G., Hüglin, C., Baltensperger, U., Szidat, S. and Prévôt, A. S.: Evaluation of the absorption Angström exponents for traffic and wood burning in the Aethalometer-based source apportionment using radiocarbon measurements of ambient aerosol, Atmospheric chemistry and physics, 17(6), 4229–4249, 2017.

---

## Author Comment (AC3) · 22 Oct 2019

First, authors would like to thank the reviewer for his/her important comments and interesting suggestions. We bieleve they clearly helped highlighting the main conclusions of the paper and extend the interest to regional impacts of aerosol sources in South-America.

As recommend by all reviewers, layers have been renamed in order to justify that the method used only allows to differentiate atmospheric dynamic. Thus, Stable Layer (SC) and Turbulent Layer (TL) are replaced by Stable Condition (SC) and Turbulent Condition (TC).

In the following, authors answer to the reviewer and list modifications made to the paper.

The paper by Aurelien et al. presents a detailed analysis of aerosol optical properties at a remote site of Andean mountains. The measurements are long-term and, therefore, certainly credible in terms of seasonal cycles, however, the novelty is mainly based on somewhat underexplored and sensitive region and that alone does not constitute scientific novelty. The authors make up for publishable results by careful and thorough data analysis separating dataset to represent stable and turbulent atmospheric layers alongside detailed trajectory analysis. The paper can be accepted for publication after addressing mainly minor comments. Last but not least English can be improved with the help of senior co-authors.

Conclusions could be more concise if a summarising diagram with the main transport patterns and corridors were presented. What matters is not a repeat of study results, but emphasising something about lasting regional impacts. Otherwise, it is just another study of optical properties at a different location.

**Answer:** Figure has been improved in order to illustrate the main airmasses transported to CHC station according seasons. The main influences for different seasons are discussed from l.274. As the RC3 suggests, this information gives interesting elements to discuss about regional impacts of the different aerosol sources in this particular region. Regional impacts have been highlighted in the final conclusion.

In addition, three tables have been added to summarize ranges of Angström exponent for the different aerosol types (Table 1), to detail the median values of the Angström exponent for each cluster, season and atmospheric stability measured at CHC station (Table 2), and suggest a new Angtsröm exponent definition for the different aerosol types (Table 3).

**Modifications:**

Figure 4 has been improved.

[Figure]

**Figure 4: b. Monthly variation of the percentage of back-trajectories (BTs) for each cluster.**

Table 1, 2 and 3 have been added.

| Aerosol type | SAE | AAE | SSAAE |
|---|---|---|---|
| Dust | Close to 1 | Close to 1 | Below 0 |
| Urban pollution | Close to 2 | Close to 1 | Higher than 0 |
| Biomass burning | | Close to 2,1 | |

Table 1: Expected aerosol type and their optical properties for each cluster according season and atmospheric stability.

| Cluster | season | SAE | AAE | SSAAE | Aerosol types |
|---|---|---|---|---|---|
| NA | WET | 2,04 (1,42) | 0,58 (0,56) | 0,18 (0,15) | urban (dust/urban) |
| | DRY | 1,91 (1,80) | 1,00 (1,01) | 0,01 (0,004) | urban (dust) |
| | BB | 1,92 (1,87) | 1,10 (1,26) | 0,03 (0,02) | dust/BB (dust/BB) |
| SA | WET | 1,2 (1,40) | 0,74 (0,68) | 0,11 (0,11) | urban (urban) |
| | DRY | 1,69 (1,70) | 1,04 (0,96) | 0,02 (0,03) | dust (dust) |
| | BB | 2,16 (2,02) | 1,23 (1,20) | 0,005 (0,01) | BB (BB) |
| LP | WET | 1,71 (2,09) | 0,86 (0,82) | 0,08 (0,10) | urban (urban) |
| | DRY | 1,64 (1,74) | 1,05 (1,07) | 0,02 (-0,01) | urban (dust/urban) |
| | BB | 1,49 (1,93) | 1,09 (1,29) | -0,02 (-0,02) | dust (dust/BB) |
| ATL | WET | 1,93 (2,11) | 0,75 (0,65) | 0,11 (0,15) | urban (urban) |
| | DRY | 1,77 (1,94) | 1,00 (1,05) | -0,001 (0,006) | dust (dust/urban) |
| | BB | 1,80 (1,81) | 1,23 (1,08) | 0,008 (0,01) | dust/BB (urban) |
| APO | WET | 2,15 (2,04) | 0,84 (0,82) | 0,11 (0,10) | urban (urban) |
| | DRY | 1,39 (1,38) | 1,06 (1,10) | 0,006 (-0,02) | dust (dust) |
| | BB | 1,56 (1,61) | 1,14 (1,20) | -0,008 (-0,01) | dust/BB (dust/BB) |
| NES | WET | 2,05 (1,67) | 0,72 (0,66) | 0,13 (0,12) | urban (urban) |
| | DRY | 1,74 (1,83) | 1,06 (1,09) | -0,008 (0,003) | dust/urban (dust) |
| | BB | 1,89 (1,80) | 0,95 (1,07) | 0,002 (0,02) | dust/urban (urban) |

Table 2: Median aerosol Angström exponents of turbulent condition (stable condition) for each cluster and seasons measured at the CHC station and resulting aerosol types.

| Aerosol type | SAE | AAE | SSAAE |
|---|---|---|---|
| Dust | - | > 0,9 | [-0,05 ; 0,05] |
| Urban pollution | > 1,4 | < 0,9 | > 0,05 |
| Biomass burning | - | > 1,1 | [-0,05 ; 0,05] |

Table 3: Updated Angström exponent values expected for aerosol types at the CHC station.

l.127: "As a summary, Table 1 shows expected Angström exponent for dust, urban pollution and Biomass Burning particules according the different referenced works (Dubovik et al., 2002 ; Collaud Coen et al., 2004 ; Clarke et al., 2007 ; Russel et al., 2010). This information has to be taken with caution since source influences are expected homogeneous and have been reported from several regions."

l.482: "Table 2 summarizes the median Angström exponents measured at the CHC station for turbulent conditions (stable conditions in parenthesis). According to these values and as discussed above, aerosol types for the turbulent conditions (and stable conditions in parenthesis) are given."

l.525: "A new Angtsröm exponent classification can then be defined for measurement at the CHC station and is reported Table 3. Thresholds are close to the ones proposed by previous works (Dubovik et al., 2002 ; Collaud Coen et al., 2004 ; Clarke et al., 2007 ; Russel et al., 2010) but adapted to CHC's instruments and particular atmospheric conditions."

l.4503: "The present study clearly demonstrates the regional impacts of these activities."

l.505: "The present study has hence demonstrated that BB particles are efficiently transported to the higher part of the troposphere (Stable conditions) and over long distances (more than 300 km long)."

l.510: "One of the main aerosol sources in the Bolivian plateau is the urban area of La Paz / El Alto."

l.520: "Finally, the arid plateau of the region has also demonstrated regional impact. In addition to urban and BB influences, the wavelength dependence of the single scattering albedo (SSAAE) measured at CHC highlights a main dust influence during the entire dry season with SSAAE values close to 0."

Minor comments

Line 31. Resulting in lower atmospheric...

**Answer:** In the final version, this sentence has been deleted.

Line 33. different aerosol sources.

**Answer:** In the final version, this sentence has been deleted.

Line 35. on average, instead of "in average".

**Answer:** Correction made.

**Modification:**

l.39: "[...] extinction coefficients are on average [...]"

Line 41. ...increase in the extinction...

**Answer:** Correction made.

**Modification:**

l.54: "28% to 80% increase in the extinction"

Line 44. How far away?

**Answer:** Scales still difficult to describe when only one in-situ station is used in addition to back-trajectories. In the present study, "long distance influences" is used for aerosol particles transported several hundreds of kilometres far from their sources whereas "local influence" is linked to aerosol particles transported less than 100 km from their sources. Additional ground based measurements could help to locate with more detail aerosol sources and satellite measurements could give more information on their long distance transport.

**Modification:**

l.46: "[...] far away from their sources [...]" is replaced by "[...] to a remote site [...]"

Is stable layer normally the upper layer above the boundary layer or is it free troposphere? I understand it is based on statistical treatment, but the abstract should convey the message without reading all the details.

**Answer:** From the method based on the local dynamic of the atmosphere and the impossibility to access to the vertical distribution of aerosol for the full period, stable and turbulent conditions are used in this study to differentiate atmospheric layers. Hence, it is not possible to attribute stable and turbulent conditions to unique common tropospheric layers. A sentence is added on the abstract to make this step clearer.

**Modification:**

l.58: "Results are also separated from distinct atmospheric conditions as stable and turbulent, with associated properties of the free troposphere and the planetary boundary layer"

Line 106-113. There have to be goals, not summary and justification of what and how the study has done.

**Answer:** The paragraph has been modified according to these suggestions.

**Modification:**

l.133: "Monthly and diurnal variations of extensive optical properties (related to particle concentration) and intensive optical properties (related to particle chemistry) are firstly shown. A robust method based on the measurement of the atmospheric stability is then applied to distinguish atmospheric conditions (stable and turbulent). Finally, back-trajectory analysis and optical wavelength dependences are presented to identify impacts of local and regional aerosol sources."

Figure 1. In insert would help to visualise in the larger region, especially tha the Figure covers only appr. 50x50km.

**Answer:** A larger view of the topography is added to figure 1.

**Modification:**

[Figure]

**Figure 1: Topographic description of La Paz and Chacaltaya region, and Bolivia in the lower right panel. The black rectangle on the small panel represents La Paz region. The urban area of La Paz-El Alto (marked as white shading) lies in the Altiplano high-plateau at around 4000 m a.s.l..**

Line 125. The papers deals with impacts over much larger region, therefore, it is important to describe that larger region, e.g. extending to 200km.

**Answer:** As discussed in previous remarks, long range transport is related to several hundreds of kilometres. Precisions have been added to the text.

Line 193. Use past tense as measurements represent the past not present day.

**Answer:** Correction made.

**Modification:**

l.250: "This 3-day example showed that [...]"

Line 276. Correlation is a scientific term, therefore, cannot "correlate to seasons". Use "...values exhibited typical seasonal variation".

**Answer:** Correction made.

**Modification:**

l.347: "[...] variations of AAE and SSAAE values exhibited typical seasonal variation."

Line 327. I am not sure I follow why SL particles are aged longer and transported farther. Please elaborate.

**Answer:** Aerosol particles reaching high altitudes are subject to a more stable atmosphere with strong horizontal circulation and weak vertical motions. Hence, especially small particles (diameters around 1 µm) can be aged longer and transported farther.

**Modification:**

l.399: "[...] SC aerosol particles are aged longer and transported farther than TC particles due to less scavenging effects."

**References:**

Clarke, A., McNaughton, C., Kapustin, V., Shinozuka, Y., Howell, S., Dibb, J., Zhou, J., Anderson, B., Brekhovskikh, V., Turner, H. and Pinkerton, M.: Biomass burning and pollution aerosol over North America: Organic components and their influence on spectral optical properties and humidification response, J. Geophys. Res. Atmospheres, 112(D12), D12S18, doi:10.1029/2006JD007777, 2007.

Collaud Coen, M., Weingartner, E., Apituley, A., Ceburnis, D., Fierz-Schmidhauser, R., Flentje, H., Henzing, J. S., Jennings, S. G., Moerman, M., Petzold, A., Schmid, O. and Baltensperger, U.: Minimizing light absorption measurement artifacts of the Aethalometer: evaluation of five correction algorithms, Atmospheric Measurement Techniques, 3(2), 457–474, doi:10.5194/amt-3-457-2010, 2010.

Dubovik, O., Holben, B., Eck, T. F., Smirnov, A., Kaufman, Y. J., King, M. D., Tanre, D. and Slutsker, I.: Variability of absorption and optical properties of key aerosol types observed in worldwide locations, J. Atmospheric Sci., 59, 590–608, doi:Review, 2002.Russell, P. B., Bergstrom, R. W., Shinozuka, Y., Clarke, A. D., DeCarlo, P. F., Jimenez, J. L., Livingston, J. M., Redemann, J., Dubovik, O. and Strawa, A.: Absorption Angstrom Exponent in AERONET and related data as an indicator of aerosol composition, Atmospheric Chem. Phys., 10, 1155–1169, doi:10.5194/acp-10-1155-2010, 2010.

Russell, P. B., Bergstrom, R. W., Shinozuka, Y., Clarke, A. D., DeCarlo, P. F., Jimenez, J. L., Livingston, J. M., Redemann, J., Dubovik, O. and Strawa, A.: Absorption Angstrom Exponent in AERONET and related data as an indicator of aerosol composition, Atmospheric Chem. Phys., 10, 1155–1169, doi:10.5194/acp-10-1155-2010, 2010.